# Axonal TDP-43 condensates drive neuromuscular junction disruption through inhibition of local synthesis of nuclear encoded mitochondrial proteins

Topaz Altman[1,9], Ariel Ionescu[1,9], Amjad Ibraheem[1], Dominik Priesmann [2], Tal Gradus-Pery[1], Luba Farberov[1], Gayster Alexandra[3], Natalia Shelestovich[3], Ruxandra Dafinca [4], Noam Shomron[1,5], Florence Rage [6], Kevin Talbot [4], Michael E. Ward[7], Amir Dori [8], Marcus Krüger [2] & Eran Perlson [1,5 ✉]

Mislocalization of the predominantly nuclear RNA/DNA binding protein, TDP-43, occurs in motor neurons of ~95% of amyotrophic lateral sclerosis (ALS) patients, but the contribution of axonal TDP-43 to this neurodegenerative disease is unclear. Here, we show TDP-43 accumulation in intra-muscular nerves from ALS patients and in axons of human iPSC-derived motor neurons of ALS patient, as well as in motor neurons and neuromuscular junctions (NMJs) of a TDP-43 mislocalization mouse model. In axons, TDP-43 is hyper-phosphorylated and promotes G3BP1-positive ribonucleoprotein (RNP) condensate assembly, consequently inhibiting local protein synthesis in distal axons and NMJs. Specifically, the axonal and synaptic levels of nuclear-encoded mitochondrial proteins are reduced. Clearance of axonal TDP-43 or dissociation of G3BP1 condensates restored local translation and resolved TDP-43-derived toxicity in both axons and NMJs. These findings support an axonal gain of function of TDP-43 in ALS, which can be targeted for therapeutic development.

[1] Sackler Faculty of Medicine, Tel-Aviv University, Tel-Aviv, Israel. [2] CECAD Research Center and Center for Molecular Medicine (CMMC), University of Cologne, 50931 Cologne, Germany. [3] Pathology Institute, Sheba Medical Center, Tel Hashomer, Ramat Gan, Israel. [4] Nuffield Department of Clinical Neurosciences, University of Oxford, Oxford, UK. [5] Sagol School of Neuroscience, Tel-Aviv University, Tel-Aviv, Israel. [6] Institut de Génétique Moléculaire de Montpellier, IGMM UMR535 Montpellier, France. [7] National Institute of Neurological Disorders and Stroke, National Institutes of Health, Bethesda, MD, USA. [8] Department of Neurology, Sheba Medical Center, Tel Hashomer and Sackler Faculty of Medicine, Tel Aviv University, Ramat Gan, Israel. [9] These authors contributed equally: Topaz Altman, Ariel Ionescu. ✉email: eranpe@tauex.tau.ac.il

Amyotrophic lateral sclerosis (ALS) is an adult-onset neurological disease characterized by neuromuscular junction (NMJ) disruption and motor neuron (MN) degeneration[1,2]. An important pathological hallmark in ALS patients is mislocalization of the primarily nuclear RNA and DNA binding protein TAR-DNA-binding-protein 43 (TDP-43) to the cytoplasm of MNs[3–7]. TDP-43 participates in transcription, RNA splicing, processing, and nucleocytoplasmic transport[4,8–10]. Additionally, mutations in TDP-43 were identified in a subset of ALS patients[11]. Cytoplasmic accumulation of TDP-43 has been implicated in ALS via several pathways[12], some are related to the loss of TDP-43 nuclear function[4,8,10]. However, the mechanism sensitizing MNs and NMJs to TDP-43 cytoplasmic condensation remains unclear.

One key event which develops due to TDP-43 mislocalization is the formation of phase separated cytoplasmic condensates that alter RNA localization and translation[13–17]. Additionally, ALS-associated mutations in TDP-43 directly interfere with mRNA transport[18,19]. This process is associated with development of pathological ribonucleoprotein (RNP) condensates, which deregulate mRNA localization and translation[20,21].

Formation of RNP condensates[21,22] and mRNA transport defects[23] affect localized protein synthesis, an important regulator of axonal and synaptic health[24–27]. Several studies demonstrated altered protein synthesis in ALS models[14,28,29]. However, most of those observations were made in non-neuronal cells or within neuronal cell body, not in the NMJ. Given that NMJs and MN axons are highly vulnerable in ALS[30,31], the consequences of TDP-43 mislocalization in these compartments are key to understanding disease pathology.

To study the effect of TDP-43 mislocalization on the NMJ in a precise and controlled environment, we used a neuromuscular co-culture setup in microfluidic chambers (MFCs) we developed[32–36]. This platform models pathological features of ALS, such as axon degeneration, NMJ dysfunction and MN death[33,35,37]. The fluidic separation between MN cell-body and axon allows formation of functional NMJs exclusively at distal compartments[32–38], enabling to study local protein synthesis events at the subcellular level.

Here, we describe a toxic gain-of-function of TDP-43 accumulation in axons and NMJs, which impacts synaptic protein synthesis and provokes neurodegeneration. We demonstrate that TDP-43 mislocalization leads to formation of axonal RNP-complexes that interfere with local protein synthesis in axons and NMJs. This process reduces the levels of nuclear-encoded mitochondrial proteins, consequently leading to NMJ dysfunction. Finally, clearance of TDP-43 or breakdown of G3BP1 condensates in axons reverses the pathological events, signifying a possible pathway for MN recovery in ALS.

## Results

### Phosphorylated TDP-43 accumulates in motor axons in ALS or upon induced TDP-43 mislocalization. TDP-43 mislocalizes to the cytoplasm in ALS patient spinal cord MNs, where it is observed in highly ubiquitinated and phosphorylated insoluble aggregate-like structures[3,4]. However, the spread of TDP-43 pathology to MN axons and NMJs was not thoroughly investigated. To determine whether TDP-43 axonal mislocalization occurs in ALS patients, we immuno-stained TDP-43 in muscle biopsies from sporadic ALS and non-ALS patients. This revealed that the levels of both TDP-43 and its pathological phosphorylated form (pTDP-43)[3,16] are elevated in intra-muscular nerves of ALS patients (Fig. 1a–d). Next, we tested the existence of axonal TDP-43 pathology in C9ORF72 ALS patient iPS-MNs[39,40], as previously reported for cell-bodies[7]. Evidently, C9ORF72 iPS-MNs had a significant increase in axonal levels of pTDP-43

compared to its isogenic control line (Fig. 1e–f, Supplementary Fig. 1). Thus, pTDP-43 accumulates in MN axons of ALS patients.

To further study the role of TDP-43 axonal mislocalization, we utilized inducible transgenic mice expressing the human TDP-43 lacking the nuclear-localization-signal (ΔNLS) through the doxycycline (dox) TET-off system. This TDPΔNLS mouse model recapitulates ALS-like MN disease pathologies[41–43] including NMJ disruption, in a mechanism that is not fully understood. To precisely monitor MNs, TDPΔNLS mice were crossbred to Choline-Acetyltransferase (ChAT)cre-tdTomatolox mice (hereafter ChATtdTomato). Retraction of dox from adult animals or from primary MN culture, resulted in TDP-43 cytoplasmic mislocalization, and to elevated TDP-43 and pTDP-43 levels in sciatic nerve motor axons and even remote NMJs (Fig. 1g–m, Supplementary Fig. 2a–b). To confirm TDP-43 mislocalization into axons, we developed a radial MFC that allows collection of pure axonal protein and RNA in large quantities. Using this system, we found an increase in axonal TDP-43 and pTDP-43 levels upon dox-retraction from MNs (Fig. 1n–p, Supplementary Fig. 2c–g). Thus, inducing TDP-43 mislocalization in TDPΔNLS mouse MNs, mimics our observations from ALS patients MN axons.

### Axonal TDP-43 colocalizes with G3BP1 and forms RNP Condensates. Upon its perinuclear mislocalization in ALS MNs, TDP-43 forms insoluble aggregate-like structures, often in its phosphorylated form[13–16]. To determine whether TDP-43 creates similar condensates in axons, we grew primary TDPΔNLS MN in compartmentalized cultures, and tested TDP-43 colocalization with the RNP component, Ras GTPase-activating protein-binding-protein1 (G3BP1)[20,21]. Evidently, TDP-43 extensively colocalizes with G3BP1 upon dox retraction in all neuronal compartments of TDPΔNLS MNs (Supplementary Fig. 3a–l). Other RNA binding proteins, such as FMRP and FUS, did not show similar colocalization with G3BP1 in TDPΔNLS MN axons (Supplementary Fig. 3m–p). Notably, using co-labeling with SYTO RNA-select dye[18], we found that RNA, a critical component of cytoplasmic RNP condensates, is enriched in the axonal TDP-43-G3BP1 complexes (Fig. 2a–c, Supplementary Fig. 3q–r). Furthermore, we observed formation of similar axonal RNP complexes between pTDP-43, G3BP1, and RNA in C9ORF72 iPS-MNs (Fig. 2d–f, Supplementary Fig. 4), and between pTDP-43 and G3BP1 in patient intra-muscular nerves (Fig. 2g–i). Thus, mislocalized TDP-43 associates with G3BP1 to form RNP condensates along MN axons.

### Axonal TDP-43 RNP assembly reduces local protein synthesis in MN axons and NMJs. Formation of G3BP1-positive RNP condensates is strongly associated with repressed RNA translation[22] and altered mRNA transport[20], leading to reduced protein synthesis[21]. We therefore sought to determine whether TDP-43 axonal accumulation, and subsequent RNP condensate formation, impair local protein synthesis in MN axons and NMJs. To test this, we cultured C9ORF72 iPS-MNs in MFCs and applied O-Propargyl-Puromycin (OPP) to the axonal compartment to label newly synthesized proteins. We found a substantial decrease in density of the OPP puncta in C9ORF72 axons (Fig. 3a–b, Supplementary Fig. 5a–c), which was less apparent in cell-bodies (Supplementary Fig. 6). This decrease in translation was reversed by axonal-exclusive application of TAT-fused peptide corresponding to residues 190-208 of G3BP1 (G3BP1 peptide), which was recently reported to dissociate G3BP1 condensates[21] (Fig. 3c–h, Supplementary Fig. 7). Thus, axonal RNP condensate

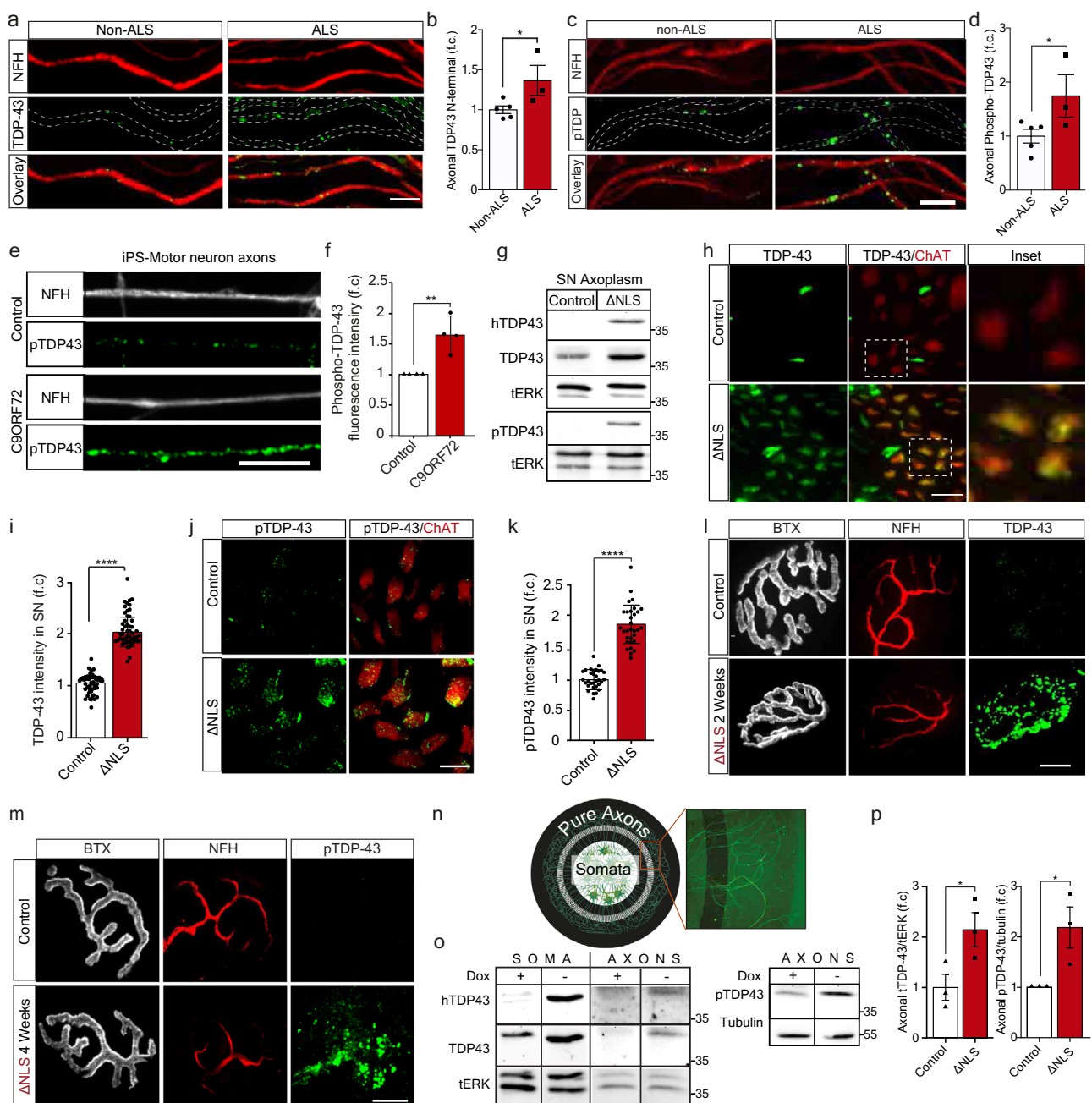

**Fig. 1 TDP-43 mislocalizes to MN axons in ALS patients and TDPΔNLS mice. a, c** Immunofluorescence images and **b, d** Quantification of non-ALS and ALS patient intra-muscular nerves TDP-43 (**b**) or phosphorylated TDP-43 (pTDP-43) (**d**) signal within NFH-positive axons, normalized to NFH intensity. Scale bar=10 μm $n = 5$, 3 patients. Data are presented as mean values ± SD. Unpaired-$t$-test, one-sided, *$P = 0.026$ (**b**),0.031 (**d**). **e, f** Images (**e**) and quantification (**f**) of pTDP-43 in axons (NFH) of C9ORF72 and control iPS-MN. Scale bar=10 μm. $n = 210,181$ axons. Data are presented as mean values ± SEM. Unpaired-$t$-test, two-sided. **$P = 0.0042$. **g** Western-blots for TDP-43, human-specific-TDP-43 (hTDP-43) and pTDP-43 in TDPΔNLS and control mice sciatic nerve axoplasm (SN axoplasm). ERK1/2 (tERK) used as loading control. $n = 3$, 3 mice. **h–k** Images (**h, j**) and quantification (**i, k**) of TDP-43 (**h**) or pTDP-43 (**j**) intensity in ChAT-positive sciatic nerve axons of TDPΔNLS[ChAT::tdTomato] and control[ChAT::tdTomato] mice. Scale bar=20 μm (**h**), 10 μm (**j**). $n$ (**h–i**)=50,48, $n$ (**j–k**)=31,31 images from 3,3 mice. Data are presented as mean values ± SD. Unpaired-$t$-test, two-sided. ****$P < 0.0001$. **l–m** Images of TDP-43 (**l**) or pTDP-43 (**m**) in NMJs before, two-weeks (**l**) and four-weeks (**m**) after dox retraction from TDPΔNLS mice diet. Bungarotoxin (BTX) and NFH mark the pre- and postsynaptic compartments respectively. Scale bar=10 μm. $n$ (**l, m**)=3,3 mice. **n** Radial MFCs structure. MNs axons (HB9::GFP) grow radially-outwards through microgrooves for large-scale protein/RNA purification. **o, p** Western-blots (**o**) and quantification (**p**) of hTDP-43, TDP-43 and tERK (left), and pTDP-43 and tubulin (right) in protein lysates of TDPΔNLS or control MNs/axons isolated from radial MFCs. $n = 3$ repeats. Data are presented as mean values ± SEM. Unaired-$t$-test, one-sided, *$P = 0.028$ (left panel), Unpaired-$t$-test, two-sided, *$P = 0.043$ (right panel). f.c stands for Fold Change. Source data are provided as a Source Data file.

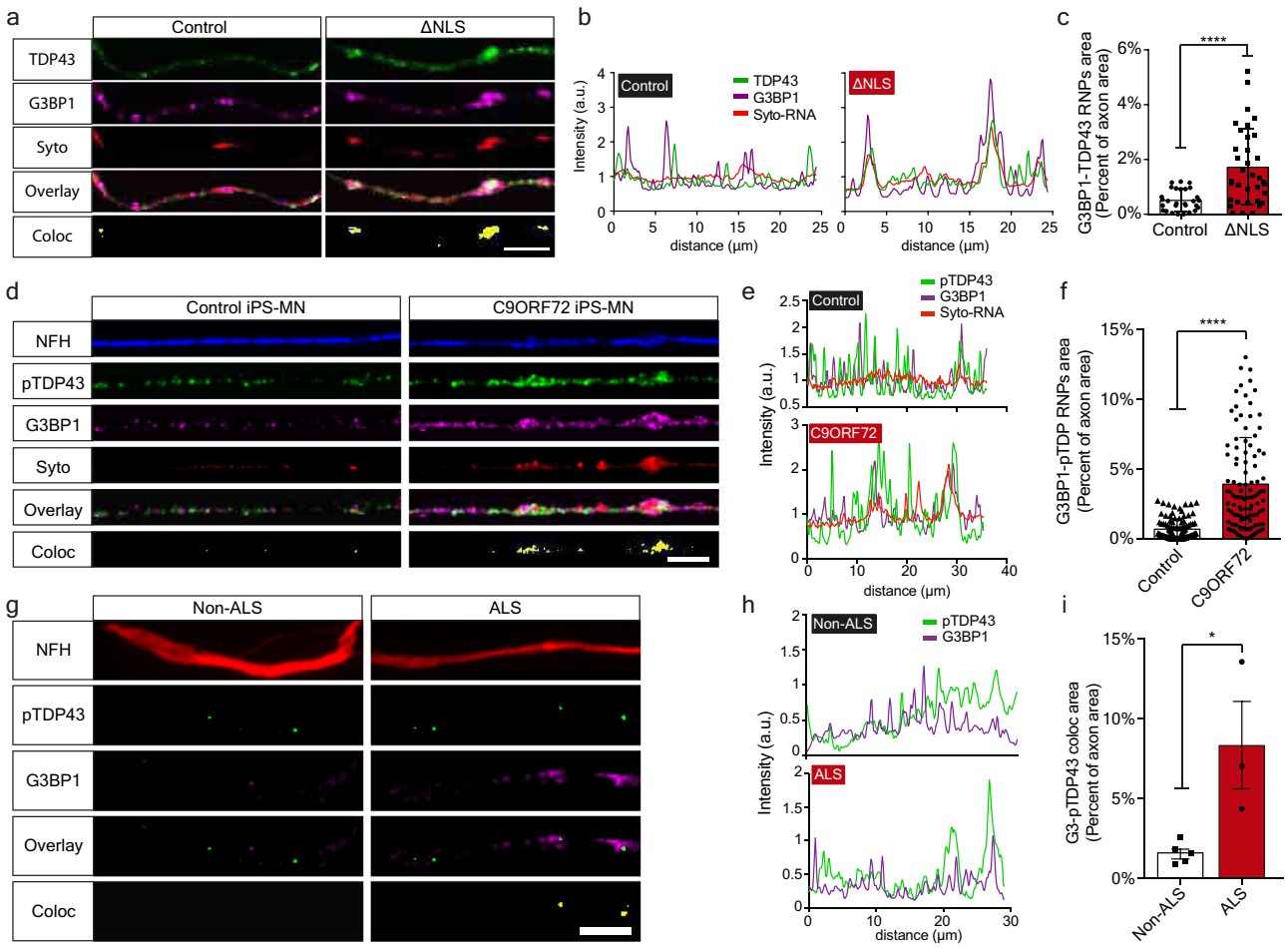

**Fig. 2 Axonal TDP-43 accumulation leads to formation of pathological RNP condensates. a–c** Images (**a**), colocalization profiles (**b**) and quantification (**c**) of TDP-43-G3BP1-SytoRNA colocalization in control or ΔNLS axons. Scale bar=10 μm. $n = 28,35$ axons. Data are presented as mean values ± SD. Unpaired-$t$-test, two-sided. ****$p < 0.0001$. **d–f** Images (**d**), colocalization profiles (**e**) and quantification (**f**) of pTDP-43-G3BP1-SytoRNA colocalization in control or C9ORF72 iPS-MN. Scale bar=10 μm. $n = 108,104$ axons. Data are presented as mean values ± SD. Unpaired-$t$-test, two-sided, ****$p < 0.0001$. **g–i** Images (**g**), colocalization profiles (**h**) and quantification (**i**) of pTDP-43-G3BP1 colocalization in non-ALS or ALS patient intra-muscular nerves. Scale bar=10 μm. $n = 5,3$ patients. Data are presented as mean values ± SEM. Unpaired-$t$-test, two-sided. *$P = 0.0163$. a.u stands for arbitrary units. Source data are provided as a Source Data file.

assembly drives suppression of local protein synthesis. To ensure that local protein synthesis inhibition occurs due to axonal TDP-43, we cultured primary TDPΔNLS MNs in MFCs, and applied OPP to the axonal compartment. Again, we observed a major decrease in axonal translation (Fig. 3i–j), similar to that seen by inducing RNP-condensates with sodium-arsenite (NaAsO₂)[44] or by inhibiting protein synthesis using Anisomycin and Cyclo-heximide, all exclusively in axons (Supplementary Fig. 5d–g). Next, we aimed to evaluate local protein synthesis also in the most remote point of the MN axon, the NMJ. To that end, we performed co-cultures of TDPΔNLS MNs with healthy puromycin-resistant muscles to avoid post-synaptic staining (Supplementary Fig. 8a–c). By quantifying OPP puncta density within in-vitro NMJs, we identified robust protein synthesis within control co-cultures, while protein synthesis in TDPΔNLS NMJs was severely impaired (Fig. 3k–l). A similar decrease in OPP density was observed after NaAsO₂ application (Supplementary Fig. 8d–e). Following these findings, we examined the extent of protein synthesis interference in adult TDPΔNLS mice. Labeling and quantification of the OPP signal in sciatic nerves and hind-limb muscles revealed a reduction in the amount of newly synthesized proteins both in sciatic MN axons and in NMJ

pre-synapse of TDPΔNLS mice (Fig. 3m–p, Supplementary Fig. 9, Supplementary Movie 1)[24]. Taken together, we demonstrate that TDP-43-induced axonal assembly of RNP condensates disrupts protein synthesis in axons and in the NMJ pre-synapse.

**TDP-43 axonal accumulation reduces the levels of nuclear-encoded mitochondrial proteins by suppressing protein translation.** To determine which proteins are primarily affected by axonal TDP-43 accumulation and reduction in local protein synthesis, we performed proteome analysis of sciatic nerve axo-plasm samples from TDPΔNLS and control mice. The analysis revealed a global reduction in nuclear-encoded mitochondrial proteins (Mitocarta 2.0[45]), including respiratory chain complex proteins (Fig. 4a–c, Supplementary Data 1). We specifically validated this reduction for two nuclear-encoded mitochondria proteins, Cox4i1 and ATP5A1 in TDPΔNLS MN axons (Fig. 4d–h, Supplementary Fig. 10a).

Since mRNAs of nuclear-encoded mitochondrial genes are among the most abundant mRNAs in MN axons[46,47] we sought to determine whether the reduction in nuclear-encoded mito-chondrial proteins occurs at the transcriptional or translational level. We conducted RT-qPCR analysis for three nuclear-encoded

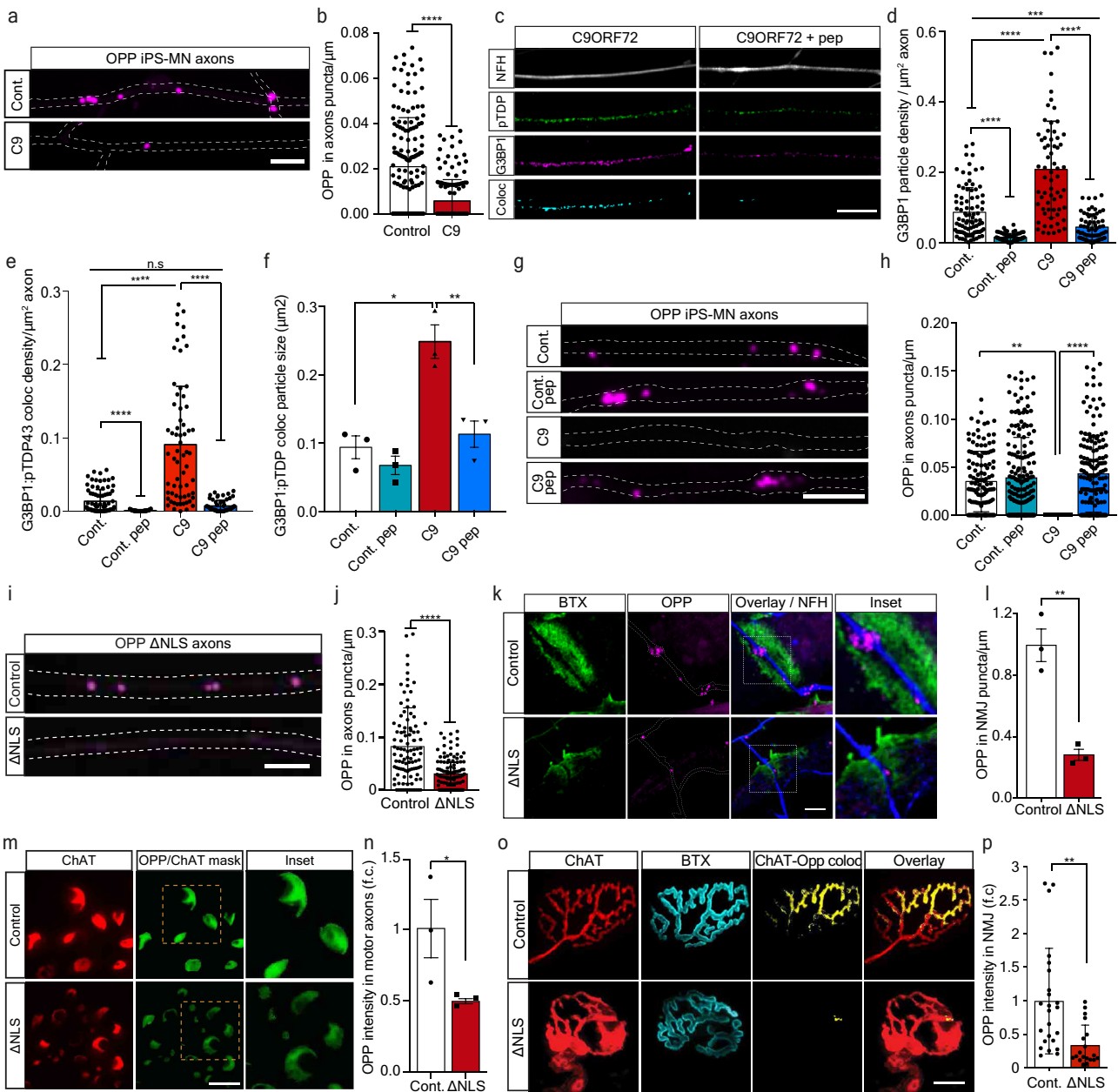

**Fig. 3 TDP-43 mediated formation of axonal condensates reduces Local Synthesis in MN axons and NMJs. a** Images and **b** quantification of OPP puncta density in C9ORF72 or control iPS-MN axons. Scale bar=5 μm. $n$ = 159,144 axons. Data are presented as mean values ± SD. Unpaired-$t$-test, two-sided ****$p$ < 0.0001. **c** Images and **d**–**f** quantification of NFH, pTDP-43 and G3BP1 and their colocalization in C9ORF72 iPS-MN axons either treated or not with G3BP1 peptide. Quantifications of G3BP1 particle density (**d**), G3BP1-pTDP-43 coloc-particle density (**e**) and G3BP1-pTDP-43 coloc-particle size (**f**) from C9ORF72 and control iPS-MN axons, either treated or not with G3BP1 peptide. Scale bar=10 μm. $n$ (**d** and **e**)=91,71,65,68 axons, $n$(**f**)=3 repeats, each over 1000 particles. Data are presented as mean values ± SD. One-way-ANOVA with Holm-Sidak correction. *$P$ = 0.0151,**$P$ = 0.0074, ***$P$ = 0.0005, ****$p$ < 0.0001. **g** Images and **h** quantification of OPP puncta density from C9ORF72 and control iPS-MN axons either treated or not with G3BP1 peptide. Scale bar=5 μm. $n$ = 130,71,193,173 axons. Data are presented as mean values ± SD. One-way-ANOVA with Holm-Sidak correction. **$P$ = 0.094, ****$p$ < 0.0001. **i** Images and **j** quantification of OPP puncta density in control or ΔNLS MN axons. Scale bar=5 μm. $n$ = 100,112 axons. Data are presented as mean values ± SD. Unpaired-$t$-test, two-sided. ****$p$ < 0.0001. **k** Images and **l** quantification of OPP puncta density in in-vitro NMJs from control or ΔNLS co-cultures. BTX indicates post-synapse, NFH indicates pre-synapse. Scale bar=5 μm. $n$ = 3 independent repeats. Data are presented as mean values ± SEM. Unpaired-$t$-test, two-sided. **$P$ = 0.0031. **m** Images and **n** quantification of OPP-labeled sciatic nerve sections from TDPΔNLS^ChAT::tdTomato or control^ChAT::tdTomato mice. Scale bar=10 μm. Data are presented as mean values ± SEM. $n$ = 3,3 mice. Unpaired-$t$-test, one-sided. *$P$ = 0.0364. **o** Images and **p** quantification of pre-synaptic OPP in NMJs from TDPΔNLS^ChAT::tdTomato or control^ChAT::tdTomato mice. Scale bar=10 μm. Data are presented as mean values ± SD. $n$ = 24,19 NMJs, from 3,3 mice. Unpaired-$t$-test, two-sided. **$P$ < 0.0044. f.c stands for Fold Change. Source data are provided as a Source Data file.

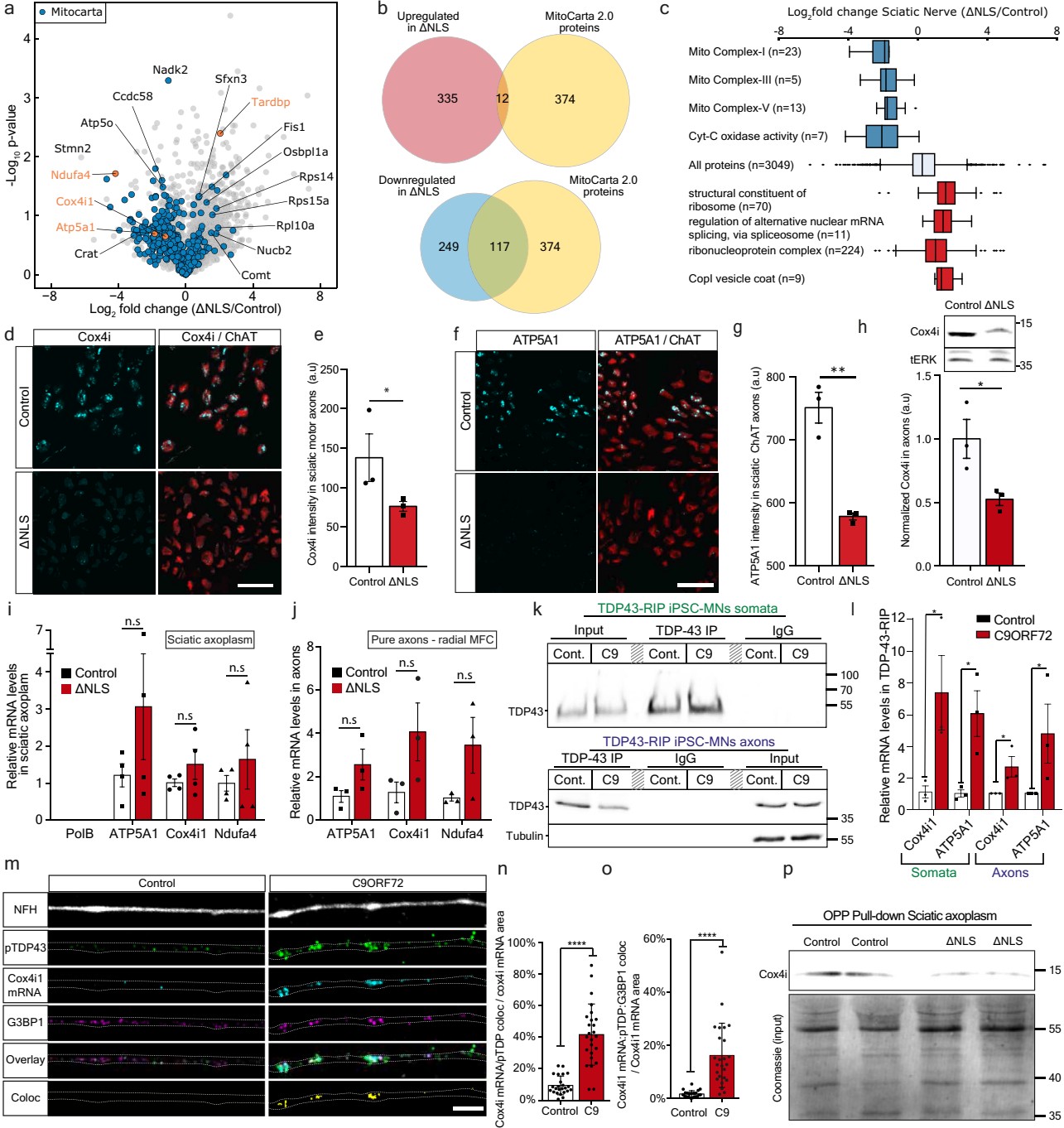

mitochondrial genes: ATP5A1, Cox4i1, and Ndufa4 in sciatic nerve axoplasm and cultured MN axons. This analysis revealed a slightly increased mRNA abundance, contradicting their reduced protein levels, suggesting that TDP-43 accumulation impairs nuclear-encoded mitochondrial proteins local translation (Fig. 4i–j, Supplementary Fig. 10b).

Next, we hypothesized that the mRNAs of these proteins are directly bound and sequestered within TDP-43 axonal RNP condensates. To assess this, we performed TDP-43 RNA-immuno-precipitation (RIP) from the somata and axonal compartments of C9ORF72 iPS-MNs cultured in radial MFCs. RT-qPCR analysis of TDP-43-bound mRNAs detected a substantial increase in the

binding of both Cox4i1 and ATP5A1 mRNAs to TDP-43 in diseased MN somata and axons (Fig. 4k–l). Additionally, smFISH for Cox4i1 mRNA in C9ORF72 iPS-MN axons combined with immunostaining for pTDP-43 and G3BP1 revealed extensive sequestration of Cox4i1 mRNA within TDP-43 positive axonal RNP condensates (Fig. 4m–o, Supplementary Fig. 10c). Finally, to test if the localization of Cox4i1 mRNA into axonal RNP condensates inhibits its translation, we performed streptavidin pull-downs of biotinylated-OPP from sciatic nerve axoplasm samples of TDPΔNLS mice[24]. Evidently, OPP bound Cox4i1 was decreased in TDPΔNLS sciatic axoplasm, demonstrating that reduction of Cox4i1 levels derives from impaired axonal translation

**Fig. 4 TDP-43 axonal accumulation reduces the levels of nuclear-eEncoded mitochondrial proteins by suppressing protein translation. a** Volcano-plot of sciatic nerve axoplasm proteome analysis from TDPΔNLS and control mice. Data is shown as Log₂FC of TDPΔNLS over control proteome. $n = 4,4$ mice. Nuclear-encoded mitochondrial proteins (Mitocarta) in blue. **b** Venn-diagram of all Mitocarta proteins that were up-regulated (upper panel) or down-regulated (lower panel). Log₂FC > 1 for up-regulated and < −1 for down-regulated. **c** GO analyses categories. Blue=down-regulated, red=up-regulated. Center of the box plots shows median values, boxes extent from 25% to the 75% percentile, whiskers show 5% and 95% percentile. n is the number of proteins as specified in the figure per category. **d−g** Images and quantification of Cox4i (**d**, **e**) and ATP5A1 (**f**, **g**) levels in ChAT-positive axons within TDPΔNLS$^{ChAT::tdTomato}$ and control sciatic nerve cross-sections. Scale bar=20 μm. $n = 3,3$ mice. Data are presented as mean values ± SEM. Mann−Whitney test, one-tailed, *$P = 0.05$, Unpaired-$t$-test, one-sided, **$p < 0.0011$. **h** Western-blot and quantification of Cox4i in isolated axons from TDPΔNLS or control MNs cultures. tERK used as loading control. $n = 3,3$ repeats. Unpaired-$t$-test, two-sided, *$p = 0.0413$. **i−j** RT-qPCR of Cox4i1, ATP5A1 and Ndufa4 mRNA in TDPΔNLS and control sciatic axoplasm (**i**) or pure axons of TDPΔNLS or control MNs cultured in radial MFC (**j**). mRNA levels normalized to mitochondria content (MT-RNR1 levels). PolB mRNA indicates no nuclear-RNA contamination. $n$(**i**)=3,3 mice, $n$(**j**)=3,3 repeats. One-way-ANOVA with Holm-Sidak correction. Data are presented as mean values ± SEM. **k, l** TDP-43-RIP of somata (upper panel) and pure axons from radial MFC (lower panel) in C9ORF72 and control iPS-MNs, immunoblotted for TDP-43 and tubulin as a loading control. **l** RT-qPCR of Cox4i1 and ATP5A1 mRNA levels following TDP-43-RIP from somata and axons. in C9ORF72 and control iPS-MNs. $n = 3$ independent repeats. Data are presented as mean values ± SEM. Mann–Whitney test, one-sided *$P = 0.05$ (somata Cox4i1), Unpaired-$t$-test, two-sided *$P = 0.0261$(somata ATP5A1), Unpaired $t$-test, one-sided, $P = 0.0275$ (axons Cox4i1), Mann–Whitney test, one-sided, $P = 0.05$. **m–o** Images (**m**) and quantification (**n**, **o**) of Cox4i1 mRNA-pTDP-43 colocalization (**n**) and Cox4i1 mRNA-pTDP-43-G3BP1 colocalization (**o**) in C9ORF72 and control iPS-MN axons. Scale bar=5 μm. $n = 24,25$ images from 3 repeats. Data are presented as mean values ± SD. Unpaired-$t$-test, two-sided ****$p < 0.0001$. **p** Immunoblot for Cox4 following OPP pull-down (upper panel) and Coomassie staining (total protein) of input lysates (lower panel) from TDPΔNLS and control sciatic axoplasms labeled with OPP. $n = 10$ mice (20 sciatic nerves) per each lane. a.u stands for arbitrary units. Source data are provided as a Source Data file.

(Fig. 4m, Supplementary Fig. 10d–e). Thus, we show that TDP-43 mislocalization leads to deficient local protein synthesis of nuclear encoded mitochondrial genes such as Cox4i1, through sequestration of their mRNA within axonal RNP condensates.

**TDP-43 mislocalization impairs axonal and NMJ mitochondria via limitation of mitochondria-related protein synthesis**. Our observations so far suggest TDP-43 mislocalization affects axonal synthesis of nuclear-encoded mitochondrial proteins through sequestration or their mRNAs in axonal RNP-condensates. Yet, the extent of damage this process projects on axonal and synaptic mitochondria remains elusive. We first assessed the effect of acute axonal protein synthesis inhibition over mitochondria activity using TMRE. Local axonal administration of both aniso-mycin and CHX, as well as NaAsO₂ led to a notable reduction in mitochondrial membrane potential (Fig. 5a–b). Thus, interfering with axonal protein synthesis impedes mitochondria function. As recent findings indicate that synthesis of nuclear-encoded mito-chondrial proteins occurs in proximity to mitochondria[25], we further examined the association of OPP with mitochondria in TDPΔNLS axons. We identified a profound reduction in this colocalization, with a similar trend to that obtained with protein synthesis inhibition (Fig. 5c–d, Supplementary Fig. 11a–b). Therefore, mitochondria associated protein synthesis in axons is decreased by TDP-43 mislocalization.

Next, we measured the direct effect of TDP-43 mislocalization on mitochondria integrity in distal axons by generating TDPΔNLS$^{Thy1-mito-Dendra}$ mice, which express mitochondria targeted florescent tag in neurons. This approach exposed an extensive reduction in axonal and pre-synaptic mitochondria upon induction of TDP-43 mislocalization (Fig. 5e–h). Further-more, by co-culturing TDPΔNLS$^{Thy1-mito-Dendra}$ primary MNs with healthy muscles we were capable of measuring TMRE signals in pre-synaptic mitochondria, that were significantly reduced in NMJs following TDP-43 mislocalization (Fig. 5i–j). Intriguingly, axonal TMRE signals in TDPΔNLS MNs cultured alone were only mildly affected by TDP-43 mislocalization and became more apparent when assessing mitochondrial recovery from acute anisomycin challenge (Supplementary Fig. 11c–e). Critically, mitochondria activity in the NMJs of TDPΔNLS$^{Thy1-mito-Dendra}$ MNs was recovered following localized administration of G3BP1 peptides to the NMJ compartment (Fig. 5i–j). Hence, axonal TDP-43 RNP-condensates impair mitochondria activity and

integrity in NMJs by limiting local translation of vital mitochon-drial components.

**Mitochondria activity and local synthesis are vital for NMJ function and their inhibition leads to neurodegeneration**. Thus far, we demonstrated that mislocalized TDP-43 forms RNP-condensates along MN axons that interfere with protein synth-esis, specifically of nuclear-encoded mitochondrial proteins. However, the functional outcome of this impairment remains unclear. We therefore tested whether NMJ activity depends on general mitochondrial health using MN infection with lentivirus encoding mitochondrial-targeted Killer-Red fusion protein (MKR)[48]. Upon NMJ formation, pre-synaptic NMJ mitochondria were exclusively irradiated, followed by live imaging of calcium transients in post-synaptic muscles (Fig. 6a–b). In response to mitochondria irradiation, we observed a marked decrease in muscle activity (Fig. 6c–d, Supplementary Movies 2–3), suggest-ing that pre-synaptic mitochondria are necessary to maintain NMJ activity.

Next, to determine if local protein synthesis has a similar contribution to NMJ function, we inhibited protein synthesis in pre-synaptic axons by applying puromycin exclusively to the NMJ compartment in co-cultures with puromycin-resistant muscles (Supplementary Fig. 8a–c). Imaging calcium transients in post-synaptic muscles revealed a pronounced decrease in active muscles following axonal and synaptic protein synthesis inhibi-tion (Fig. 6e–h, Supplementary Movies 4–5). These results indicate that local protein synthesis is fundamental to maintain active NMJs.

Finally, having found that both mitochondria and local protein synthesis are essential for NMJ activity, we aimed to determine if MN axons exhibit neurodegeneration in response to protein synthesis inhibition. An evaluation of axon health over time revealed that axonal application of protein synthesis inhibitors leads to extensive degeneration (Supplementary Fig. 12). Furthermore, TDPΔNLS axons had increased sensi-tivity to protein synthesis inhibition (Fig. 6i–j). Altogether, we show that both mitochondrial function and local protein synthesis play crucial roles in maintaining axonal integrity and NMJ function. Both processes are consequently impaired following TDP-43 mislocalization, possibly leading to NMJ dysfunction and neurodegeneration.

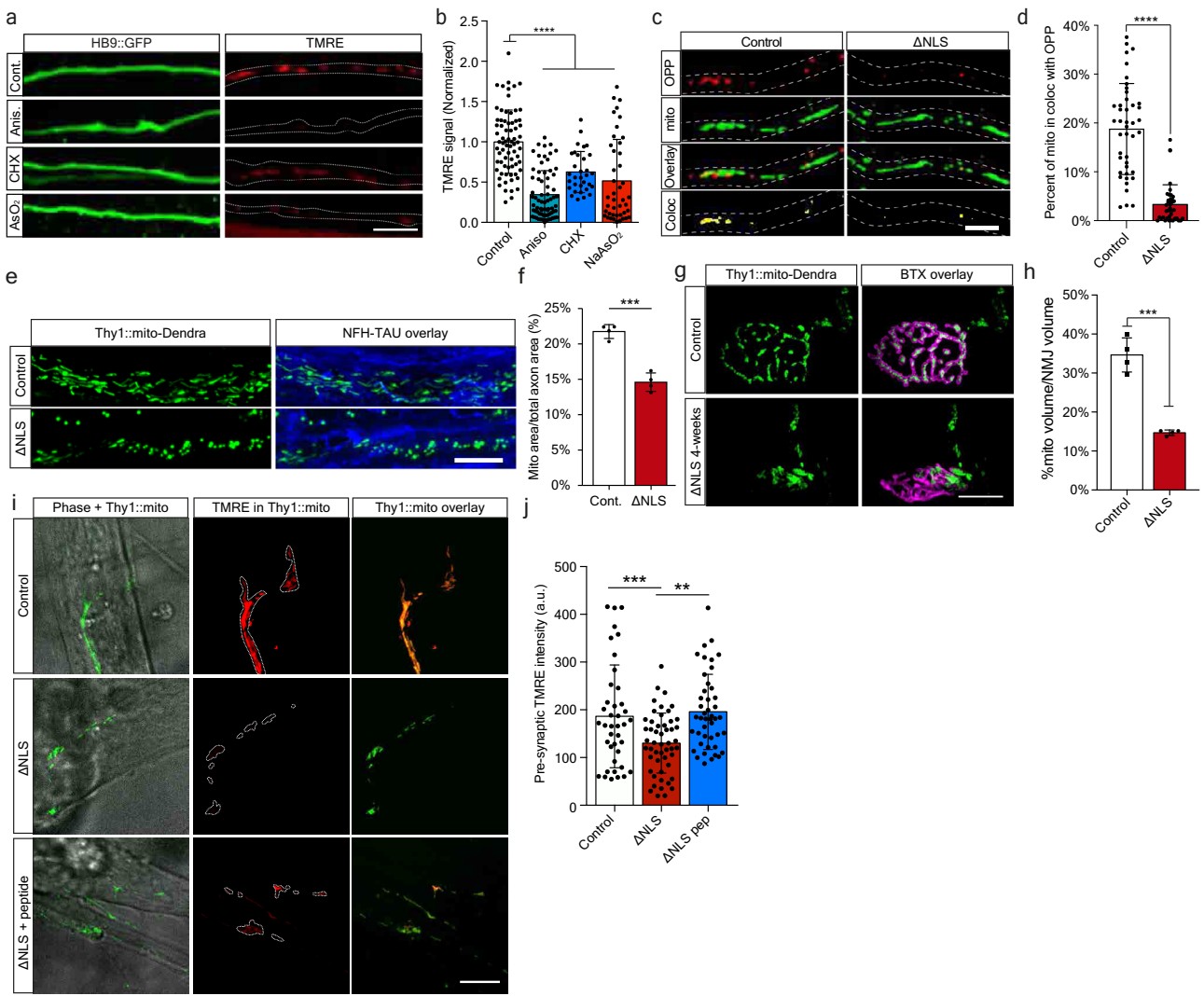

**Fig. 5 TDP-43 mislocalization impairs axonal and NMJ mitochondria via limitation of mitochondria-related protein synthesis. a** Images and **b** quantification of TMRE signal in HB9::GFP MN axons untreated (control) or treated with anisomycin (aniso), cycloheximide (CHX) or NaAsO2. Scale bar=10 μm. $n$ = 72,69,43,32 axons. Data are presented as mean values ± SD. One-way-ANOVA with Holm-Sidak correction. ****$p$ < 0.0001. **c** Images and **d** quantification of OPP and mitochondria colocalization in TDPΔNLS or control MN axons. Scale bar=5 μm. $n$ = 43,36 axons. Data are presented as mean values ± SD. Unpaired-$t$-test, two-sided ****$p$ < 0.0001. **e** images and **f** quantification of mitochondria density within sciatic nerve longitudinal sections of TDPΔNLS^Thy1::MitoDendra and control mice labeled also for NFH and Tau. Scale bar=10 μm. $n$ = 4,4 mice. Data are presented as mean values ± SEM. Unpaired-$t$-test, two-sided. ***$p$ = 0.0001. **g** Images and **h** quantification of pre-synaptic mitochondrial volume within NMJs of TDPΔNLS^Thy1::MitoDendra and control^Thy1::MitoDendra mice. BTX marks post-synapse. Scale bar=10 μm. $n$ = 4,4 mice. Data are presented as mean values ± SEM. Unpaired-$t$-test, two-sided. ***$p$ = 0.0001. **i** Images and **j** quantification of pre-synaptic TMRE signal intensity within in vitro NMJs from TDPΔNLS^Thy1::MitoDendra, control^Thy1::MitoDendra co-cultures and TDPΔNLS^Thy1::MitoDendra co-cultures treated with G3BP1 peptides in NMJ compartment. Scale bar=10 μm. $n$ = 40,51,45 NMJs. One-way-ANOVA with Holm-Sidak correction. Data are presented as mean values ± SD. **$p$ = 0.0034, ***$p$ = 0.0005. a.u stands for arbitrary units. Source data are provided as a Source Data file.

**Clearance of TDP-43 condensates recovers local translation of nuclear-encoded mitochondrial proteins and enables NMJ reinnervation.** After gaining mechanistic understanding of the pathological outcome of axonal TDP-43 condensate formation on local translation and mitochondria health, we examined whether clearance of TDP-43 condensates can reverse this harmful process. To study the effect of axonal TDP-43 clearance, as previously performed in MNs cell bodies[43], we investigated the ability of TDPΔNLS mice to recover after ceasing the expression of TDPΔNLS and allowing endogenous TDP-43 redistribution to normal localization. We employed a recovery paradigm in which doxycycline was reintroduced into the diet of TDPΔNLS mice after initial deprivation (same was done in vitro, see methods).

Re-introduction of doxycycline lowered TDP-43 and pTDP-43 levels in the spinal cord and allowed partial re-distribution into nuclei (Supplementary Fig. 13a–d). Evidently, recovery also had impact on the levels of hTDP-43 in sciatic axoplasm, that was accompanied by a reduction in the overall TDP-43 levels (Fig. 7a–c, Supplementary Fig. 13a). Most importantly, TDP-43 was cleared from NMJs (Fig. 7d–e). Furthermore, quantification of the OPP signal in the pre-synaptic side of NMJs upon clearance of TDP-43 revealed that protein synthesis in MN axons returns to full capacity, both in-vitro (Fig. 7f–g) and in-vivo (Fig. 7h–i, Supplementary Fig. 13e–f). We also performed colocalization analysis of OPP signal with mitochondrial proteins Cox4i and ATP5A1 in hind-limb muscles, which implied that

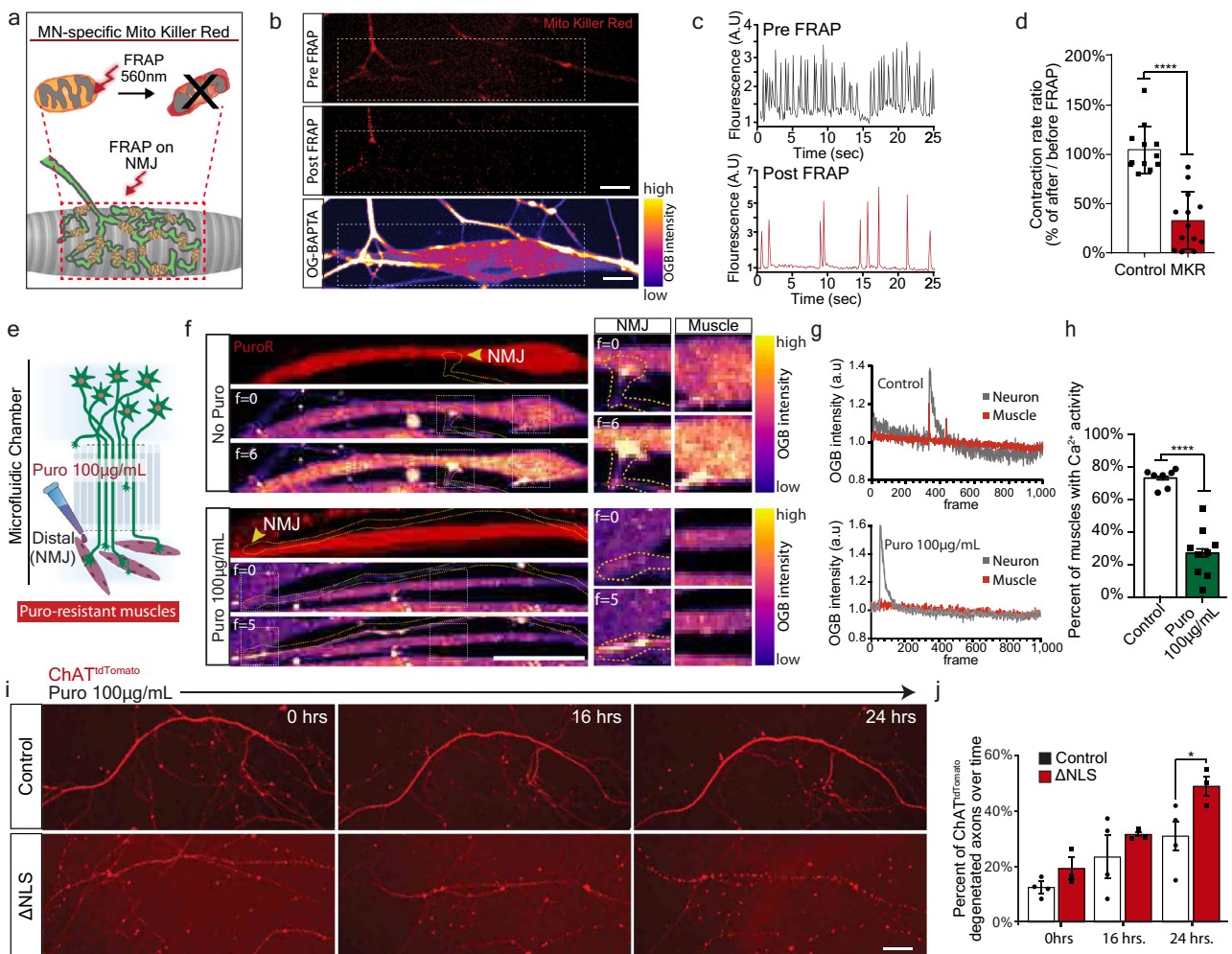

**Fig. 6 NMJ function is dependent on mitochondria and local synthesis, and their absence leads to decreased NMJ activity and axon degeneration. a** Schematic illustration of Mito-Killer-Red (MKR) experimental setup, used for specifically targeting oxidative stress to NMJ mitochondria. **b** Images of MKR in NMJ pre-synapse before and after bleach (white line=bleached region). Lower panel indicates post-synaptic muscle labeled with calcium indicator Oregon-Green-BAPTA (OGB). Scale bar=10 μm. n = 13,12 NMJs from 3 independent experiments. **c**, **d** Representative OGB time-trace of OGB indicating muscle contraction (**c**) and quantification (**d**) of contraction ratio before and after MKR bleaching. As control, ChAT::tdTomato-expressing axons were bleached instead of MKR-expressing axons. n = 13,12 NMJs from 3 independent experiments. Data are presented as mean values ± SD. Unpaired-t-test, two-sided. ****p < 0.0001. **e** Schematic illustration of experimental procedure for puromycin local protein synthesis inhibition in NMJ pre-synapse using puromycin resistant muscles. **f** Left panel: Time-series images of OGB-labeled co-cultures treated, or not with puromycin in NMJ compartment. Right panel: Demonstration of paired axon-muscle calcium activity only in control NMJs and its absence upon puromycin application. Scale bar=20 μm. n = 7,9 MFCs from 3 independent experiments. **g** Time traces of OGB in pre-synaptic neurons and post-synaptic muscles in control (upper plot), and in puromycin-treated (lower plot) cultures. **h** Quantification of the percent of innervated and contracting muscles after puromycin application. n = 7,9 MFCs from 3 independent experiments. Data are presented as mean values ± SD. Unpaired-t-test, two-sided. ****p < 0.0001. **i** Images and **j** quantification of the percent of degenerating axons in TDPΔNLS and control MN cultures following 16 and 24 h of puromycin treatment. Scale bar=50 μm. n = 4,3 MFCs from 3 independent experiments. Data are presented as mean values ± SD. Two-way-ANOVA with Holm-Sidak correction. *p = 0.0407. a.u stands for arbitrary units. Source data are provided as a Source Data file.

both proteins undergo local synthesis in NMJs of control mice, but to a lesser extent in TDPΔNLS NMJs (Fig. 7j–m, Supplementary Fig. 13g–j, Supplementary Movie 6). Therefore, clearance of TDP-43 from NMJs lead to recovery of pre-synaptic Cox4i and ATP5A1.

Finally, we investigated the functional impact of TDP-43 mislocalization on NMJ degeneration, and whether it could be reverted by applying the recovery paradigm. Strikingly, measurement of the innervation rate in TDPΔNLS NMJs in-vivo and in-vitro revealed that mislocalized TDP-43 facilitates NMJ dysfunction and disruption both in adult mice and in co-culture, a process which is reversed by restoring TDP-43 localization (Fig. 8a–d, Supplementary

Fig. 13k–n). Furthermore, aside of axonal regeneration and NMJ reinnervation, TDP-43 mislocalization also resulted in lack of functional NMJ activity, as measured by the percent of innervated muscles that contract. Importantly, this was reversed either by restoring TDP-43 localization with dox re-introduction (Fig. 8e) or by locally dissociating RNP-condensates with G3BP1 peptides (Fig. 8f). Thus, TDP-43 derived axon and NMJ degeneration can be reverted either by direct clearance of axonal TDP-43 or by breakdown of TDP-43 RNP-condensates.

Altogether, we show that the pathological mislocalization of TDP-43 in MN axons disrupts axonal and synaptic protein synthesis. This leads to altered mitochondrial protein turnover in

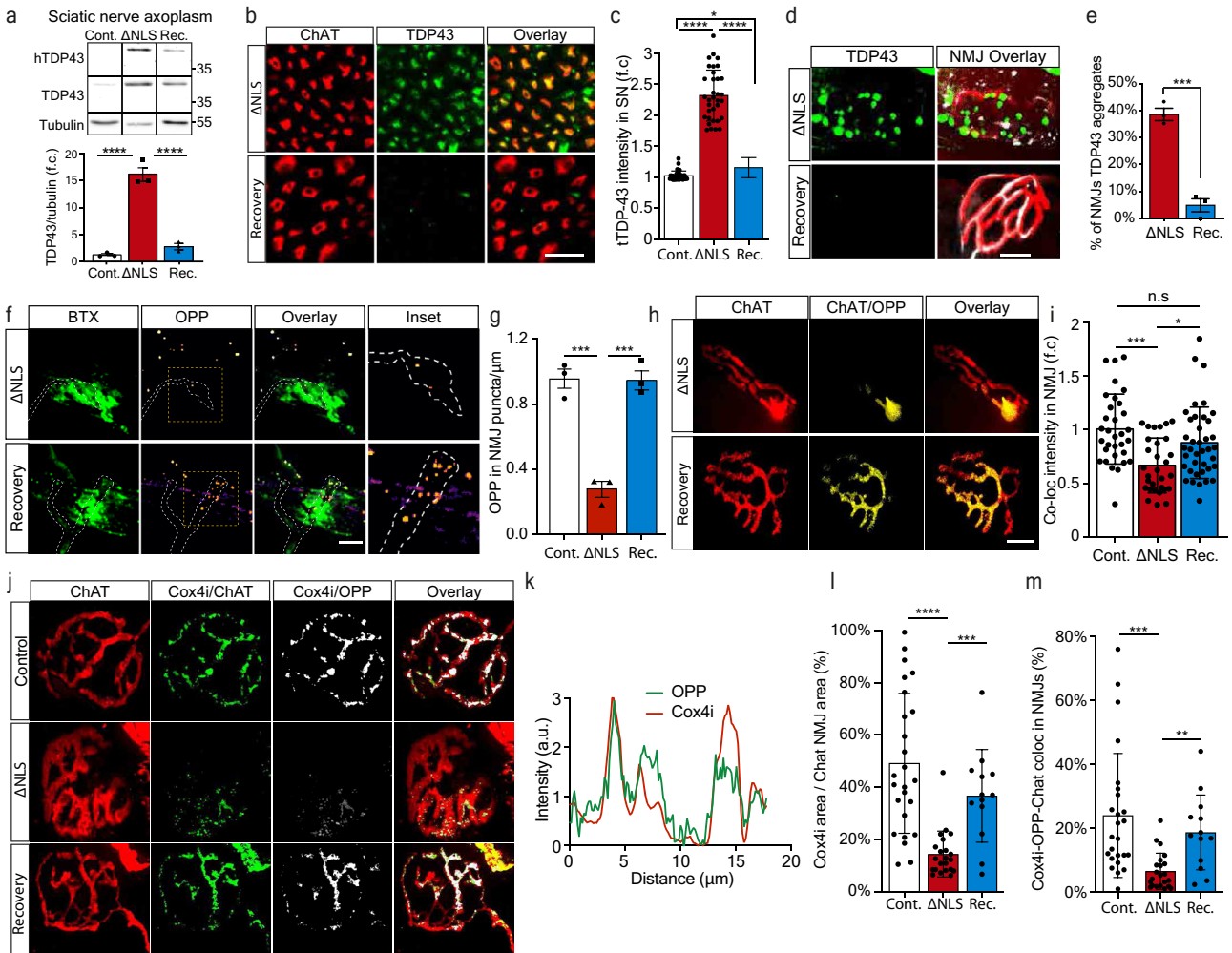

**Fig. 7 Restoring TDP-43 localization recovers local translation of mitochondrial proteins in distal axons and the NMJs. a** Western-blot and quantification of TDP-43, hTDP43 levels in sciatic axoplasm of control, TDPΔNLS, and recovered-TDPΔNLS (Rec.). Tubulin was used as loading control. n = 3,3,3 mice. Data are presented as mean values ± SEM. One-way-ANOVA with Holm-Sidak correction. ****p < 0.0001. **b** Images and **c** quantification of TDP-43 intensity within ChAT-positive axons in sciatic nerve sections of control, TDPΔNLS, and recovered-TDPΔNLS mice. Scale bar=20 μm. n = 34,34,33 images from 3 mice of each condition. Data are presented as mean values ± SD. One-way-ANOVA with Holm-Sidak correction *p = 0.0412, ****p < 0.0001. **d** Images and **e** quantification of the percent of NMJs with apparent TDP-43 condensates in control, TDPΔNLS, and recovered-TDPΔNLS mice. Scale bar=10 μm. n = 3,3,3 mice from each condition. Data are presented as mean values ± SEM. Unpaired-t-test, two-sided ***p = 0.0001. **f** Images and **g** analysis of OPP puncta density in in-vitro NMJs of control, TDPΔNLS and Recovered-TDPΔNLS co-cultures. Scale bar=5 μm. n = 3 independent repeats. Data are presented as mean values + SEM. One-way-ANOVA with Holm-Sidak correction. ***p = 0.0004 (left), ***p = 0.0004 (right). **h** Images and **i** analysis of the OPP labeling intensity in pre-synaptic NMJs of control, TDPΔNLS and recovered-TDPΔNLS mice. n = 32,30,39 NMJs, from 3,3,3 mice. Scale bar=10 μm. Data are presented as mean values ± SD. One-way-ANOVA with Holm-Sidak correction. *p = 0.019, ***p = 0.0001. **j** Images and **k** representative channel histograms of Cox4i and OPP intensities within pre-synaptic axon (ChAT) in NMJs of control TDPΔNLS mice, **l–m** quantification of Cox4i area (**l**), and of Cox4i-OPP coloc-area within pre-synaptic axons (ChAT) in NMJs of control, TDPΔNLS, and recovered-TDPΔNLS mice. Scale bar=10 μm. n = 23,21,13 NMJs. 3,3,3 mice. Data are presented as mean values ± SD. One-way-ANOVA with Holm-Sidak correction. ****p < 0.0001, ***p = 0.0007 (**l**), 0.0002 (**m**), **p = 0.0027. a.u stands for arbitrary units. f.c stands for Fold Change. Source data are provided as a Source Data file.

axons and in the NMJ, and eventually sensitizes the entire synapse to degeneration (Fig. 8g), a process which is reversible upon clearance of TDP-43 RNP condensates.

## Discussion

TDP-43 cytoplasmic mislocalization is a pathological hallmark of ALS, in both sporadic and familial cases[3–5], including patients with C9ORF72 mutation[7]. Previous research on TDP-43 focused on outcomes of cytoplasmic mislocalization in MN cell bodies[12,15]. Nonetheless, TDP-43 is regularly found in axons[49], where it serves a role in shuttling and localization of mRNAs[18]. Recent reports also revealed that TDP-43 is important for proper

axonal protein synthesis[50,51]. Here, we demonstrate that ALS patients display increased TDP-43 levels in intramuscular nerves, suggesting a forward propagation of TDP-43 to axons. We validate our findings using ALS patient-derived MNs from C9ORF72 mutated iPSCs, and an inducible mouse model that mimics cytoplasmic mislocalization of TDP-43. We show that TDP-43 axonal accumulation elicits the formation of RNA and G3BP1 containing RNP-condensates in axons, consequently interfering with axonal and pre-synaptic protein synthesis. Our observations indicate that TDP-43 specifically binds and sequesters nuclear-encoded mitochondrial mRNAs, thus depleting their protein levels in axons. As we show, mitochondria-related protein synthesis is essential to maintain the axon and the NMJ, and interference with

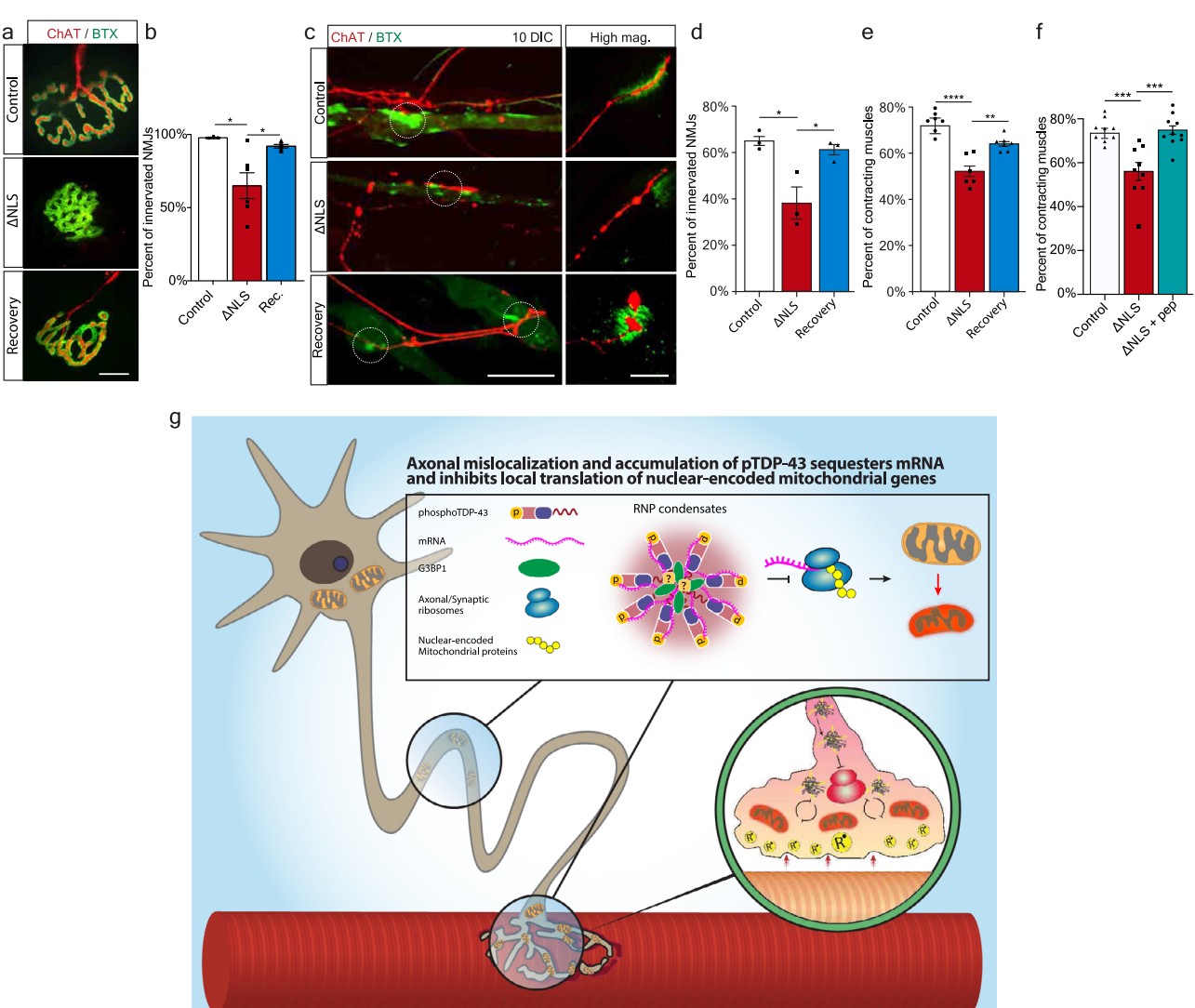

**Fig. 8 Restoring TDP-43 Localization or Dissociation of TDP-43 Condensates Enables NMJ Functional Re-innervation. a** Images and **b** Quantification of NMJs from control, TDPΔNLS, and recovered-TDPΔNLS mice. Quantification is of the percentage of innervated (Chat-BTX positive) NMJs in control, TDPΔNLS and recovered-TDPΔNLS mice. Scale bar=10 μm. $n = 3,3,3$ mice. Data are presented as mean values ± SEM. One-way ANOVA with Holm-Sidak correction *$p = 0.291$ (left), *$p = 0.291$ (right). **c** Images and **d** quantification of NMJ innervation in vitro, assessed by co-expression of pre- and post-synaptic elements (percent of ChAT-BTX clusters) in control, TDPΔNLS and recovered-TDPΔNLS co-cultures. Scale bar=40 μm in main image, 5 μm in inset. $n = 3,3,3$ MFCs. One-way-ANOVA with Holm-Sidak correction. *$p = 0.0174$(left), $0.0214$(right). **e**, **f** Quantification of the percent of contracting muscles in control, TDPΔNLS and recovered-TDPΔNLS co-cultures (**e**) or in control, TDPΔNLS and TDPΔNLS co-cultures treated with G3BP1 peptides in NMJ compartment (**f**). $n = 7,6,7$ (**e**), $10,9,9$ (**f**) MFCs. Data are presented as mean values ± SEM. One-way-ANOVA with Holm-Sidak correction. **$p < 0.0039$ (**e**), ***$p = 0.0004$ (**f**-left), $0.0003$ (**f**-right), ****$p < 0.0001$. **g** Model: axonal mislocalization of TDP-43 and its phosphorylated form create G3BP1 positive RNP condensates, which decrease local translation of nuclear-encoded mitochondrial proteins by sequestering their mRNA. This results in lack of nuclear-encoded mitochondrial proteins which leads to mitochondria toxicity and NMJ degeneration. Source data are provided as a Source Data file.

protein synthesis leads to neurodegeneration. Finally, we demonstrate that inhibition of protein synthesis is reversible, even by local restriction with RNP granule assembly, underlining the origin of this pathology in axons and further providing findings regarding the mechanisms by which MN can cope with temporary insult to their local protein synthesis capacities and to mitochondrial alterations.

The concept of local protein synthesis in neuronal processes has mainly been studied in regenerative contexts[22,24,27], but recent data also suggests it is critical for understanding neurodegenerative disease mechanisms[25,28,52]. Our findings highlight how increased abundance of TDP-43, which is hypothesized to play an important role in local protein synthesis[50,51], can become harmful upon axonal accumulation. This is associated with TDP-

43 induced formation of phase-separated cytoplasmic RNP accumulations[13,15,53]. Recently, G3BP1 positive RNP granules were shown to inhibit translation[21]. We show that translation inhibition is strongly implicated upon accumulation of axonal pTDP-43 within G3BP1-positive RNP condensates (Fig. 2, Fig. 3). Furthermore, locally dissolving G3BP1 axonal condensates restores local protein synthesis events (Fig. 3c–h). Future work will be needed to further analyze the mechanisms through which protein synthesis is regulated by formation of axonal RNP condensates.

A fundamental finding in this research is that local translation in NMJs is performed to a greater extent than in axons (i.e OPP puncta density in Fig. 3h vs. 3j). As NMJs are enriched with mitochondria[34,54,55], and possibly due to mitochondrial

dependency on local protein synthesis[25,56] (Fig. 5a–d), the enhanced protein synthesis in NMJs can be attributed to its mitochondrial density. This suggests that the high polarization of MNs leads to higher dependency on local protein synthesis of mitochondrial proteins. As we show, NMJs rely on mitochondria activity (Fig. 6a–d)[34] and on local protein synthesis (Fig. 6e–g). Therefore, interference of synaptic protein synthesis might initiate local energy deficiency that ultimately leads to NMJ degeneration. Our findings can supply an explanation for how TDP-43-mediated reduction in local protein synthesis specifically sensitizes the NMJs to rapid degeneration, and why NMJ degeneration is an early pathology in ALS[30,57]. Further research will be needed to reveal the sequence of events, focusing on the initiation of mitochondrial toxicity in the NMJ.

Finally, we found that the TDP-43 cytoplasmic-related pathology could be reversed by ceasing TDP-43ΔNLS expression, leading to reduction in axonal TDP-43 and clearance of synaptic condensates. Importantly, administration of peptides that interfere with G3BP1-TDP-43 RNP condensation seem to yield promising outcomes as well. This is a pivotal finding since in most ALS patients TDP-43 is not mutated, and yet is still mislocalized into the cytoplasm and forms aggregate-like structures. Hence, ceasing TDP-43 cytoplasmic mislocalization or dissociating TDP-43 RNP-condensates might reverse the disease outcome for a considerable number of patients, and become an important target for future drug development.

## Methods

**Transgenic mice**. NEFH-tTA line 8 (Jax Stock No: 025397) and B6;C3-Tg(tetO-TARDBP*)4Vle/J (Jax Stock No: 014650) were obtained from Jackson Laboratories and cross-bred to create NEFH-hTDP-43ΔNLS (TDP43ΔNLS) mice line. Those mice were constitutively fed with doxycycline containing diet (200 mg/kg Dox Diet #3888, Bio-Serv) as was done previously[41–43]. ChAT^cre-tdTomato^lox - hTDP-43ΔNLS was obtained by crossing ChAT^cre and tdTomato^lox mice. (Jax stock no. 006410 and 007908, respectively). Thy1-mito-Dendra- TDP-43ΔNLS was obtained by crossing the TDP-43ΔNLS with Thy1-COX8A/Dendra (Jax stock no. 025401).

After Dox retraction, all TDP-43ΔNLS mice were weighted weekly to track disease progression.

HB9-GFP (Jax stock no. 005029) mice were originally obtained from Jackson Laboratories. The colony was maintained by breeding with ICR mice (Institute of Animal Science, Harlan).

SOD1G93A (Jax stock No. 002726) mice were originally obtained from Jackson Laboratories and maintained by breeding with C57BL/6 J mice. Only non-transgenic (C57BL/6 J WT) females from this colony were used for the purpose of primary myocyte culture.

All animal experiments were approved and supervised by the Animal Ethics Committee of Tel-Aviv University.

**ALS patient-derived induced pluripotent stem-cell MNs**. Skin fibroblasts were obtained from an ALS patient carrying the (G4C2)n repeat expansion mutation in the C9ORF72 gene. Genetic confirmation, iPSC production, design of homology-directed repair templated and production of isogenic iPSC control line were performed by Prof. Kevin Talbot (University of Oxford)[39].

MN transcription factor cassette including the transcription factors Islet-1 (ISL1) and LIM Homeobox 3 (LHX3) along with NGN2 were integrated into a safe-harbor locus in iPSCs under a doxycycline-inducible promoter by Prof. Michael Ward (NIH)[40].

iPSC clones were cultured in 6-well plates coated with Matrigel (Corning; 356234), grown in mTesr1 medium (STEMCELL Technologies; 85850) and passaged with mTesr1 medium containing 10 μM Rho-Kinase Inhibitor (RI) (Sigma-Aldrich; Y0503) for 1 day following passaging. Culture media was refreshed daily until colonies reached 80% confluence. Doxycycline-induced differentiation into lower MNs was performed according to the protocol in[40] with minor modifications. Briefly, the ~300,000 iPSCs were plated in a 35 mm dish in mTesr1-RI medium. On the following day media was replaced with IM supplement containing DMEM/F12, (Gibco; 31330038), 1% N2-supplement (Gibco; 17502048), 1% NEAA (Biological Industries; #01-340-1B), 1% GlutaMAX (Gibco; 35050038) with 10 μM RI, 2 μg/mL doxycycline (Sigma, D9891), and 0.2 μM Compound E (Merck; 565790). After 48 h cells were resuspended with Accutase (Sigma-Aldrich; SCR005) and re-plated in the proximal compartment of MFCs at a concentration of 200,000 MN per MFC. Prior to plating, MFCs were coated overnight with 0.1 mg/mL PDL (Sigma-Aldrich; P6407) in PBS, and 15 μg/ml Laminin (Sigma-Aldrich; L2020) for 4 h on the following day. To prevent outgrowth of mitotically active cells 40 μM BrdU (Sigma, B9285) was added to the medium during the first 24 h after plating. At the 4th day, cells were treated with MM medium containing:

Neurobasal medium (Gibco; 21103049), 1% B27 (Gibco; 17504044), 1% N2, 1% NEAA, 1% Optimal-Culture-One supplement (Gibco; A3320201), 1 μg/mL Laminin, 20 ng/mL BDNF, 20 ng/mL GDNF, 10 ng/mL NT3 (Alomone labs; N-260). Medium was refreshed other day. For incucyte experiments 10,000 cells per well were plated in 96-well plate. Incucyte experiments were performed at 6DIV and 9DIV. Immunostaining, smFISH, OPP and RIP experiments were performed between 9DIV and 12DIV. The DIV count represents the number of days of Doxycycline-induced differentiation.

**Human muscle biopsy for intra-muscular nerve staining**. Intra-muscular nerve staining was performed on muscle biopsies from 3 ALS patients and 5 non-ALS patients (see table below). All clinical and muscle biopsy materials used in this study were obtained with written informed consent during 2016-2020 for diagnostic purposes followed by research application, approved by the Helsinki institutional review board of Sheba Medical Center, Ramat-Gan, Israel. Deltoid, quadriceps or gastrocnemius skeletal muscle samples were excised via open biopsies and pathological analysis was performed at the neuromuscular pathology laboratory at Sheba Medical Center, Ramat-Gan, Israel. All 3 ALS patients were diagnosed with clinically definite or probable ALS according to El Escorial criteria[58]. Control muscles included a variation of findings, which were consistent with a diagnosis of normal muscle, severe, chronic ongoing denervation and reinnervation due to spinal stenosis, necrotic autoimmune myopathy, type 2 fiber atrophy due to disuse and overlap myositis syndrome.

Frozen muscle biopsies were cryo-sectioned to 10 μm thick slices, mounted onto slides and air dried for 30 min in room temperature (RT). Sections were washed in PBS, fixed in 4% PFA for 20 min, and permeabilized with 0.1% Triton, and blocked with 5% goat serum (Jackson Laboratories) and 1 mg/mL BSA (Amresco). Sections were then incubated with appropriate antibodies overnight at 4 °C in blocking solution [rabbit anti TDP43 or rabbit anti phosphorylated TDP-43 (both 1:1,000, Proteintech), Chicken anti NFH (Abcam, 1:1,000). Sections were washed again and incubated for 2 h with secondary antibodies (1:1000, goat anti chicken 488 and goat anti rabbit 640, Abcam. 1:1000, goat anti mouse 594, Invitrogen), washed and mounted with ProLong Gold (Life Technologies).

**Patient details**.

| | Sex | Age | Biopsy site | Underlying clinical condition |
|---|---|---|---|---|
| ALS | M | 54 | Deltoid | ALS (sporadic) |
| | F | 33 | Quadriceps | ALS (sporadic) |
| | M | 75 | Deltoid | ALS (sporadic) |
| Non-ALS | F | 51 | Deltoid | With no evident neuromuscular condition |
| | M | 66 | Gastrocnemius | Spinal Stenosis |
| | F | 25 | Deltoid | Necrotizing Autoimmune Myopathy |
| | F | 25 | Quadriceps | Small Fiber Neuropathy |
| | F | 81 | Deltoid | Overlap Myositis/anti-synthetase Syndrome |

**Microfluidic chamber preparation**. Our MFCs design was recently published[59]. Briefly, PDMS (Dow Corning) was casted into custom-made epoxy replica molds, left to cure overnight at 70 °C, punched (6 mm/7 mm punches), cleaned and positioned in 35 mm or 50 mm glass plates (WPI): All experiments were done in 6mm-well small MFCs, except for experiments in Fig. 2, where 7mm-wells MFC were punched for spinal cord explant culture[59].

Radial PDMS molds (Fig. 1g–j) were designed and fabricated with SU-8 photoresist protocol[60] in the Tel-Aviv University Nano and Micro Fabrication Center. as described in the following table:

**Radial microfluidic chamber properties**.

| Elements | Dimensions |
|---|---|
| Inner well diameter | 7.5 mm |
| Inner well punch diameter | 7 mm |
| Inner channel height | 140 μm |
| Inner channel width | 250 μm |
| Outer channel diameter | 13 mm |
| Outer compartment punch diameter | 9 mm |
| Outer channel width | 200 μm |
| Outer channel height | 140 μm |
| Microgroove width | 10 μm |
| Microgroove length | 300 μm |
| Microgroove height | 5 μm |
| Microgroove spacing | 15 μm |
| Pod dimensions | 450 μm x 300 μm |

PDMS mold was pre-treated with Chlorotrimethylsilane (Sigma) prior to PDMS casting. PDMS casting was done in a similar manner as for regular MFC. Radial MFCs were then punched twice to form MFC rings. Inner well was punched with 7 mm biopsy punch, and outer well was punched with 9 mm punch. Cleaning procedure done in a similar manner as for regular MFC. Radial MFC rings were adhered to sterile 13 mm coverslips inside 24-well plates.

**MN culture and MN—myocyte co-culture**. E12.5 old embryos ventral spinal cord were dissected in HBSS prior to dissociation. For TDPΔNLS cultures, genotype of each embryo was determined at this phase by PCR. Meanwhile, spinal cords were kept in 37 °C 5% $CO_2$ with Leibovitz L-15 medium (Biological industries) supplemented with 5% Fetal Calf serum and 1% Penicillin/Strepto-mycin (P/S- Biological Industries). Spinal cord explants were cut transversely to small pieces and plated in MFC proximal compartment in Neurobasal (Gibco), 2% B27 (Thermo Fisher), 1% Glutamax (Gibco),1% P/S, 25 ng/mL BDNF (Alomone labs).

Dissociated MN cultures were obtained by further, trypsinization and trituration of explants. Supernatant was collected and centrifuged through BSA (Sigma) cushion. The pellet was then resuspended and centrifuged through and Optiprep (Sigma) gradient (containing 10.4% Optiprep, 10 mM Tricine, 4%w/v glucose). MN-enriched fraction was collected from the interphase, resuspended and plated in the proximal MFC compartment at a concentration of 150,000 MN per regular MFC, 250,000 per radial MFC. MNs were maintained in complete neurobasal (CNB) medium containing Neurobasal, 4% B27, 2% horse serum (Biological Industries), 1% Glutamax, 1% P/S, 25 μM Beta-Mercapto ethanol, 25 ng/mL BDNF, 1 ng/mL GDNF (Alomone) and 0.5 ng/mL CNTF (Alomone). Glial cell proliferation was restricted by addition of 1 μM Cytosine Arabinoside (ARA-C; Sigma) to culture medium in 1-3DIV. At 3DIV BDNF concentration in proximal compartment was reduced (1 ng/mL), while medium in distal compartment was enriched with GDNF and BDNF (25 ng/mL) to direct axonal growth.

Myocyte culture was performed as previously described[32,36]. Briefly, GC muscles from a P60 adult C57BL/6 J mouse were extracted into DMEM with 2.5% P/S/N (Biological Industries) with 2 mg/mL collagenase-I (Sigma) for 3 h, then dissociated and incubated with BioAmf 2.0 (BA; Biological Industries) in Matrigel (BD Corning) coated plates for 3 days. Myoblasts were purified by performing pre-plating for 3 consecutive days, and then plated at a density of 75,000 in small MFC, and 150,000 for large MFC (for SC explant experiments).

Muscles in co-culture were kept in BA medium for 7 days. To aid NMJ formation, media in all compartments was then replaced to poor neurobasal (PNB) medium, which contained only 1% P/S and 1% Glutamax. Doxycycline was applied to TDP-43ΔNLS cultures only at the proximal MN compartment, in a concentration of 0.1 μg/mL, immediately after plating and throughout the entire experimental timeline. For TDPΔNLS recovery experiments (Figs. 7–8), Doxycline was applied at the proximal MN compartment after 5 DIV, and the cultures were grown until 11 DIV. For G3BP1 peptide treatment (Fig. 5I-J, Fig. 8F), 20 μM G3BP1 peptides were added exclusively to the distal compartment, starting after 7 DIV. Experiment

**Protein extraction**. Spinal cord and GC muscles were extracted from adult mice and homogenized in ice-cold PBS lysis buffer containing 1% Triton and protease inhibitors (Roche). Sciatic axoplasm was obtained from both sciatic nerves from every mouse. Sciatic nerves were sectioned and axoplasm was extracted into 100 μL PBS and protease inhibitors by gentle pressing the sections.

Extraction of axonal and somatic proteins from radial MFCs was performed as follows: Axons were extracted by first filling the inner well with high volume of PBS and applying 40 μL of RIPA lysis buffer (1% Triton, 0.1% SDS, 25 mM Tris-HCl (pH 8.0), 150 mM NaCl) to the outer compartment for 1 min. We used the same 40 μL to collect the axon lysate from two additional wells – in total, 3 wells per sample. Protein extraction from the inner (soma) compartment was then performed by replacing the PBS with 100 μL of RIPA buffer, 1-minute incubation, and then scraping of the cells. Tissue/culture lysates were centrifuged at 10,000 G for 10 min at 4 °C. Protein concentration was determined using Bradford assay (BioRad).

**Western blotting**. Protein samples were mixed with SDS sample buffer and boiled at 100 °C for 10 min, and then loaded to 10% acrylamide gels for SDS-PAGE. Proteins were transferred to nitrocellulose membranes in buffer containing 20% MeOH. Membranes were blocked with 5% skim-milk (BD) or 5% BSA for 1 h, followed by overnight incubation at 4 °C with primary antibodies: mouse anti human TDP-43 (Proteintech, 1:4000), rabbit anti TDP-43 (Proteintech, 1:2000), rabbit anti ERK1/2 (tERK; Sigma, 1:10,000), mouse anti alpha-tubulin (Abcam, 1:5,000), rabbit anti MAP2 (Milipore, 1:1000), mouse anti TAU-5 (Abcam, 1:250). Membranes were then washed with TBST and incubated 2 h at RT with secondary HRP antibody (Donkey anti rabbit and donkey anti mouse, Jackson Laboratories, 1:10,000), or HRP-Goat-IgG2a-anti-mouse (Jackson, 1:20,000; for puromycin blots) washed with TBST and visualized in iBright 1500 ECL imager (Life Technologies) after 5 min incubation with ECL reagents. Quantification was performed using FIJI ImageJ V.2.0.0 software.

**RNA extraction and cDNA synthesis**. MN Axonal RNA was extracted from outer compartment of radial MFCs at 14DIV. Axonal RNA was extracted by removing the PBS (from prior wash) from the outer compartment and adding 100 μL TRI reagent lysis reagent (Sigma-Aldrich). Inner well was filled with higher volume of PBS to disable the inward flow of lysis reagent towards the inner (soma) compartment and prevent soma contamination. Axons washed off the plate by pipetting the TRI reagent around the outer well for 30 s. RNA from somata in the inner compartment was extracted with 100 μL TRI reagent, and lysate was collected in a similar manner. cDNA for axon and soma was prepared with High-Capacity Reverse Transcription Kit (Thermo; Cat. 4368814).

For sciatic nerve RNA extraction, sciatic axoplasm was obtained from 2 adult mice sciatic nerves in a tube containing 100 μL PBS and protease inhibitors, cut into small pieces and gently squeezed on ice. The axoplasm was then centrifuged at 10,000 G for 10 min at 4 °C. RNA was extracted using the RNAeasy micro kit (Qiagen) according to manufacturer's protocols.

**PCR and RT-qPCR**. Reverse Transcription was performed with High-capacity Reverse Transcription cDNA kit using random primers (Thermo Fisher Scientific). Standard PCR was done to test radial chambers axonal purity using KAPA ReadyMix using the following primers:

**Standard PCR primers**.

| Gene | forward primer | Reverse primer |
| --- | --- | --- |
| PolB | CCAAGGACAGGAGTGAATGAC | AAGCACAGAGAAGAGGCAATC |
| ACTB | GTATGGAATCCTGTGGCATC | AAGCACTTGCGGTGCACGAT |

qRT-PCR of sciatic axoplasm was done for the following genes: PolB, mitochondrial-RNR1, Cox4i, ATP5A1 and NDUFA4. Mitochondrial-RNR1 gene was used as a reference gene when calculating ΔCT, as we aim to quantify relative mRNA levels of nuclear-encoded mitochondrial genes as a part of total axonal mitochondria.

**qPCR primers**.

| Gene | Forward primer | Reverse primer |
| --- | --- | --- |
| PolB | CTACAGTCTGTGGCAGTTTCA | TGGCTGTTTGCTGGATTCT |
| Mito-RNR1 | AAACTCAAAGGACTTGGCGGT ACTTTATAT | ATACCTTTTTAGGGTTTGCTGAAGATGG |
| Cox4i | CATTTCTACTTCGGTGTGCCTTCG | GATCAGCGTAAGTGGGGAAAGCAT |
| ATP5A1 | TGTCGGATCTGCTGCCCAAAC | ACGCACACCACGACTCAAGAGC |
| Ndufa4 | CGTATTTATTGGAGCAGGGGGTACTG | GGCTCTGGGTTGTTCTTTCTGTCC |
| hCox4i | CAATTTCCACCTCTGTGTGTGTACGAG | GGCAAGGGGTGGTCACGC |
| hATP5A1 | AGTCGTGGCGTGCGTCAACTGA | CCTTACACCCGCATAGATAACAGCC |

qPCR Sybr-green reactions were performed with PerfeCTa SYBR green FastMix (QuanaBio) in a StepOne Real-Time PCR system (Thermo Fisher Scientific).

**Immunofluorescent staining for cryosections**. Sciatic nerve and spinal cord sections were prepared from fixating respective tissues in 4% PFA for 16 h at 4 °C, then incubation with 20% sucrose for 16 h at 4 °C, and cryo-embedding in Tissue-Tek OCT compound (Scigen). Tissues were then cryo-sectioned to 10 μm thick slices, washed with PBS, followed by permeabilized and blocking in solution containing 10% goat serum, 1 mg/mL BSA and 0.1% Triton in PBS for 1 h. Later the sections were incubated overnight at 4 °C with primary antibody rabbit anti TDP-43 (Proteintech, 1:2,000), rabbit anti pTDP-43 (Proteintech, 1:2,000), chicken anti NFH (Abcam, 1:500), rabbit anti Cox4i1 (Abcam, 1:500), rabbit anti ATP5A1 (Abcam, 1:500). This was followed by 2 h incubation at RT with secondary anti-body (Goat anti chicken 405, Abcam, 1:500. Goat anti chicken 488, Goat anti rabbit 640, Abcam, 1:1000. Goat anti rabbit 488, Invitrogen, 1:1000. Goat anti rabbit 594, Jackson laboratory, 1:1,000), wash with PBS and mounting with ProLong Gold (Life Technologies) containing DAPI nuclear staining.

**Whole mount NMJ staining**. Gastrocnemius (GC), Tibialis Anterior (TA) or Extensor Digitorum Longus (EDL) muscles were dissected from adult mice, cleared from connective tissue and kept in 4% PFA until use. Muscles were washed in PBS, stained for post synaptic AChR with αBTX-Atto-633 (Alomone labs) or αBTX-

TMR-594 (Sigma) at 1 µg/mL for 15 min. Next, muscles were permeabilized with ice-cold MeOH at −20 °C for 5 min, blocked and further permeabilized with 20 mg/mL BSA and 0.4% Triton for 1 h. Muscle preparations were agitated overnight at RT with appropriate antibodies: chicken anti Neurofilament heavy-chain (NFH) (1:500, abcam), rabbit anti NFH (1:500, Sigma), rabbit anti TDP-43 (proteintech, 1:2,000), rabbit anti pTDP-43 (Proteintech, 1:2,000), rabbit anti Cox4i (1:500, Abcam), rabbit anti ATP5A1 (1:500, Abcam). Next, muscles were incubated with secondary antibodies (Goat anti chicken 405, Abcam, 1:500. Goat anti chicken 488, Abcam, 1:1000. Goat anti rabbit 488, Invitrogen, 1:1000. Goat anti rabbit 594, Jackson laboratory, 1:1,000). Finally, muscles were cut to small vertical pieces and mounted with VectaShield (Vector Laboratories). Cover slides were sealed until use with nail polish.

**Immunofluorescent staining for MNs.** 10DIV TDPΔNLS MNs cultures in MFCs were fixated for 20 min in 4% PFA. For NaAsO₂ experiment, NaAsO₂ (250 µM) was applied for 1 h prior to fixation. MNs were, permeabilized with 0.1% Triton for 30 min, and then blocked for 1 h with 10% goat serum, 1 mg/mL BSA and 0.1% Triton in PBS for 1 h. Primary antibodies rabbit anti TDP-43 (Proteintech, 1:2,000), mouse anti Puromycin (Millipore, 1:1,000). Antibodies were diluted in blocking solution, and incubated with samples overnight at 4 °C. Secondary antibodies (Goat anti chicken 405, Abcam, 1:500. Goat anti chicken 488, goat anti rabbit 640, Abcam, 1:1000. Goat anti rabbit 488, Invitrogen, 1:1000. Goat anti rabbit 594, Jackson laboratory, 1:1,000) were diluted in blocking solution and incubated with samples for 2 h incubation at RT. MN Samples were mounted with ProLong Gold DAPI anti-fade reagent(Life Technologies).

**Fluorescence microscopy and image analysis.** Confocal images were captured using Nikon Ti microscope equipped with a Yokogawa CSU X-1 spinning disc and an Andor iXon897 EMCCD camera controlled by Andor IQ3 software. Phase-contrast movies of muscle contraction were acquired using the same microscope in Epi-mode and images were captured with an Andor Neo sCMOS camera. All live imaging experiments were performed with 5% CO2 and 37 °C humidified using in-situ microscope setup. Image analysis was performed using FIJI ImageJ V.2.0.0 and Bitplane Imaris 8.4.3 software.

**Co-culture contraction analysis.** NMJ activity was assessed by quantifying the percent of innervated and contracting myocytes in co-culture as previously described[36]. Briefly, after 12 days in co-culture, phase-contrast time lapse image series were acquired at a frame rate of 25 frames-per-second (25 FPS) using a X20 air objective. During imaging, cultures were maintained in controlled temperature and CO₂ environment. For obtaining contraction time traces, we used the "Time Series Analyzer V3" plugin for FIJI ImageJ V.2.0.0, and marked a region of interest on a small, high contrast and mobile region on the muscle, and then obtained the mean values for each time point.

**Turbofect transfection of puromycin resistant muscles.** Primary muscle cells were transfected with either PLKO.1 or with PQCXIP-mCherry empty backbone vectors containing Puromycin-N-acetyltansferase (PAC) gene. 150,000 primary myoblasts plated per well were in a matrigel pre-coated 24-well plate. After 4 h, myoblasts in each well were transfected with 1 µg of DNA (PLKO.1/PQCXIP-mCherry) and 4 µL Turbofect transfection reagent (Thermo scientific) prepared in a serum-free medium. Cultures were incubated with transfection reagent for 12 h in an antibiotic-free BA medium, and then washed with fresh BA medium with 1% P/S. After 4 h, cultures lifted from the 24-well plated with trypsin-C and plated in the distal compartment MFCs.

**OPP labeling of MN culture.** OPP was used to label ribosome-nascent polypeptide chains in MN cell bodies, MN axons and in neuromuscular co-cultures. OPP stock (20 mM; Life Technologies) was diluted in the appropriate medium to a final concentration of 20 µM, and then applied to either proximal/distal compartments of the MFCs to label cell bodies or axons/NMJs, respectively. Cultures were incubated with OPP for 30 min, that was chosen as the preferred time point (Sup.Fig. 5), while the opposite compartment was maintained with higher medium volume to prevent OPP flow and unspecific labeling. Anisomycin (40 µM; Sigma), cycloheximide (150 µM; Sigma), or G3BP1-inhibiting 190-208 peptide (20 µM) were applied for 30 min before OPP was added to cultures, and then together with OPP (total 1 h) for validating specific labeling of newly synthesized proteins. Cultures were then washed twice rapidly with cold PBS and fixed with 4% PFA for 15 min at RT. Co-cultures were labeled with 0.5 µg/mL αBTX-Atto-633 (alomone) for 15 min. Cultures were permeabilized with 0.1% Triton in PBS for 30 min at RT. ClickIT reaction with either Alexa-488 Picolyl-Azide or Alexa-594 Picolyl-Azide were performed following the protocols supplied by the manufacturer. Cultures were mounted with ProLong Gold Antifade Reagent. Tat-fused G3BP1-inhibiting peptide (190-208) was a kind gift from the laboratory of Prof. Jeffery L. Twiss.

**OPP labeling ex-vivo.** OPP was used to label protein synthesis in freshly dissected TA/EDL muscles and sciatic nerves. Immediately after mice were euthanized, tissues were extracted into 95% O₂ and 5% CO₂ oxygenized-ringer solution containing OPP (20 µM). TA/EDL were separated and further dissected into thinner sections prior incubation with OPP. Tissues were incubated with OPP for 35 min in 37 °C, then washed 3 times with PBS on an orbital shaker and fixed in 4% PFA for 12 h at 4 °C. Sciatic nerves were further incubated with 20% sucrose for additional 12–24 h at 4 °C, and then embedded in Tissue-Tek OCT compound. ClickIT procedure was performed only on 10 µm slices sectioned from the first/last 300 µm of sciatic nerves. Sciatic nerve sections were collected to Hitobond+ slides (Marienfeld). Sciatic nerve sections were permeabilized with 0.1% triton for 30 min followed by 3 PBS washes.

TA/EDL muscles were labeled with 2 µg/mL αBTX-Atto-633 (alomone) for 15 min and permeabilized with ice-cold MeOH for 5 min at −20 °C. Muscles were further permeabilized with 0.4% triton in PBS for 1 h at RT.

ClickIT reaction with either Alexa-488 Picolyl-Azide was performed following the protocols supplied by the manufacturer. Stained sciatic nerve sections were then mounted with ProLong Gold Antifade Reagent. Muscles were either mounted with VectaShield, or proceeded with to standard immunofluorescence labeling protocol.

**OPP biotinylation and streptavidin pull down from sciatic nerves.** OPP biotinylation and pull downs with streptaviding was performed as previously described by Terenzio et al. [24], with mild modifications. Briefly, for the purpose of pull-down and immunoblot for Cox4i, sciatic nerves from 10 mice per condition in each repeat were extracted and cut into 3-4 mm fragments. OPP Labeling was performed in DMEM + 10% FBS + 1% penicillin/streptomycin with 100 µg/ml OPP for 1 h at 37 °C. Axoplasm was extracted in PBS with 1X protease inhibitors mix and Samples were centrifuged for 10 min at 15,000 × g at 4 °C to remove sciatic fragments and cell debris. SDS added to 1% and OPP-tagged proteins were conjugated to biotin by click chemistry with 100 µM biotin-PEG3-azide, 1 mM TCEP, 100 µM TBTA and 1 mM CuSO4, for 2 h at room temperature on a rotator. After click conjugation, proteins were precipitated using 5 vol ice-cold acetone overnight at −20 °C. Pellets were washed twice in 1 mL methanol with sonication, resuspended in 1% SDS-PBS and desalted with Zeba-spin 0.5 mL 7 K cutoff columns. Protein concentrations were measured by BCA assay, 40 µg of each sample was taken for total coomassie staining, and equal amounts of protein (1.2 mg) were used for streptavidin pulldown which was carried out overnight at 4 °C in 1% NP40, 0.1% SDS in PBS and 1X protease inhibitor mix, with 80 µl of streptavidin magnetic beads (Pierce). After overnight incubation, beads were washed twice for 10 min with 1% NP40, 0.1% SDS and 1X protease inhibitor mix in PBS at room temperature, 3 times for 30 min with 6 M ice-cold urea in PBS with 0.1% NP-40 at 4 °C, and once again for 10 min with 1% NP40, 0.1% SDS and 1X protease inhibitor mix in PBS at room temperature. Proteins were eluted by boiling the beads with 5X sample buffer. PVDF blots blocked with 5% BSA in TBS-T and were incubated overnight with Cox4i antibody (Rabbit; 1:1,000) in blocking solution at 4 °C, and then for 2 h at room temperature with HRP-conjugated anti-rabbit antibody.

Specificity and sensitivity of OPP pull-down procedure also was validated in cultured HEK293T cells (CRL-3216, ATCC) similar to the procedure above, with the addition of the following controls: No OPP (1 hr at 37 °C) or anisomycin (200 µg/ml) for 2 h at 37 °C, followed by co-incubation with OPP (100 µg/ml) for 1 h. Pull-down PVDF blots were blocked with 5% BSA in TBS-T, and then incubated for 1 h with HRP-conjugated streptavidin (1:10,000) in blocking solution. Total blots were either labeled with Coomassie or in a similar manner with HRP-streptavidin.

OPP labeling and biotinylation for sciatic nerve total controls was performed from sciatic nerves of 3 mice per condition, and similar to described above for HEK-293T cells.

**Preparation of mass spectrometry samples.** For preparation of mass spectrometry samples, 30 µg axoplasmic protein lysates from WT and TDPΔNLS animals in 10% SDS buffer were precipitated in 80% acetone at -20 °C overnight. Samples were centrifuged at 18,000 × g at 4 °C for 10 min. Protein pellets were washed twice with 80% acetone, air-dried for 10 min and reconstituted in 50 µL urea buffer (6 M urea, 2 M thiourea in 10 mM HEPES/KOH, pH 8.0). Samples were reduced and alkylated with 5 mM TCEP and 20 mM CAA at RT for 30 min, followed by digestion with 0.5 µg endoproteinase Lys-C (Wako) at RT for 3 h. Samples were diluted fourfold with 50 mM ammonium bicarbonate (ABC) buffer and digested with 0.5 µg trypsin (Sigma) at RT overnight. Digestion was stopped by adding 1% formic acid and peptides were desalted using Stop-and-Go extraction tips as previously described[61].

**LC–MS/MS analysis and data processing.** Proteome analyses were performed using an Easy nLC 1000 ultra-high performance liquid chromatography (UHPLC) coupled to a QExactive Plus mass spectrometer (Thermo Fisher Scientific) with the same settings as described before[62]. Acquired MS spectra were correlated to the mouse FASTA databased using MaxQuant (v. 1.5.3.8) and its implemented Andromeda search engine[63] with all parameters set to default. N-terminal acetylation and methionine oxidation were set as variable modifications and cysteine carbamidomethylation was included as a fixed modification. Missing value imputation, statistical analyses and GO annotations were performed in Perseus (v. 1.6.2.3)58 and data were visualized in Instant Clue[64]. Significance cutoff was set to a log2 fold change of at least ±0.58 and a -log10 p-value of 1.3.

**Single-molecule fluorescent in-situ hybridization combined with immunofluorescent staining**. Labeling of single mRNA molecules in iPS-MN was performed by smiFISH as previously described[65]. Briefly, ALS Patient-derived C9ORF72 and isogenic control iPS-MN were grown for 12 days in MFCs placed over 22 mm × 22 mm coverslips. Labeling procedures were performed in RNase-free environment and using RNAse-free reagents. Cultures were fixed with 4% PFA and permeabilized overnight with 70% Ethanol. Samples were incubated with SSC (Sigma; S6639) based 15% Formamide (Thermo; 17899) buffer for 15 min. Samples were hybridized overnight at 37 °C with 13 x FLAP-Y-Cy-3 tagged complementary oligonucleotide probes targeting regions in human Cox4i1 mRNA (IDT). Hybridization mix: 15% Formamide, 1.7% tRNA (Sigma; R1753), 2% FLAP:Probe mix, 1% VRC (Sigma; R3380), 1% BSA (Roche; 10711454001), 20% Dextran Sulfate (Sigma; D8906), 1X SSC. Samples were washed twice with warm 15% Formamide, X1 SSC buffer (1 h each), and then 30 min with 1X SSC buffer, and 30 min with 1X PBS. Prior to immunostaining samples were washed with TRIS-HCl (pH = 7.5), 0.15 M NaCl buffer, and then permeabilized with same buffer with 0.1% triton. Samples blocked for 30 min with TRIS-HCl-NaCl buffer with 0.1% Triton and 2% BSA. Samples were incubated with primary antibodies in blocking buffer overnight at 4 °C, and then with fluorescent secondary antibodies in blocking buffer for 2 h at room temperatures. Samples were mounted with Vectashield Antifade reagent.

**smFISH probe sequences**.

| Probe name | Probe sequence including FLAP-Y sequence |
| --- | --- |
| hCox4i1-1 | TTCCAGTAAATAGGCATGGAGTTGCATGGCTTACACTCGG ACCTCGTCGACATGCATT |
| hCox4i1-2 | CATAGTGCTTCTGCCACATGATAACGAGCTTACACTCGGA CCTCGTCGACATGCATT |
| hCox4i1-3 | TGTTCATCTCAGCAAA GCTCTCCTTGAACTTACACTCGGA CCTCGTCGACATGCATT |
| hCox4i1-4 | GCAAGGGGTGGT CACGCCGATCCATATTACACTCGGACC TCGTCGACATGCATT |
| hCox4i1-5 | TGGAAATTGCTCGCTTGCCAACTAGGCTTACACTCGGACC TC GTCGACATGCATT |
| hCox4i1-6 | AATACCCTGGTAGCCAACATTCTGCCTTACACTCGGACCT CGTCGACATGCATT |
| hCox4i1-7 | CAGCATCTCTCACTTCTTCCACTCGTTTACACTCGGACCT CGTCGACATGCATT |
| hCox4i1-8 | TTTTCGTAGTCCCACTTGGAGGCTAAGCCTTACACTCGGA CCTCGTCGACATGCATT |
| hCox4i1-9 | GGATGGGGTTCACCTTCATGTCCAGCTTACACTCGGACCT CGTCGACATGCATT |
| hCox4i1-10 | CTGCTTGGCCACCCACTCTTTGTCAATTACACTCGGACCT CGTCGACATGCATT |
| hCox4i1-11 | ACAACCGTCTTCCACTCGTTCGAGCCTTACACTCGGACCT CGTCGACATGCATT |
| hCox4i1-12 | CAGGAGGCCTTCCTTCTCCTTCAATTACACTCGGACCT CGTCGACATGCATT |
| hCox4i1-13 | CCTTCTGGCTGGCAGACAGGTGCTTGTTACACTCGGACC TCGTCGACATGCATT |

**RNA Immunoprecipitation**. RNA-IP was performed as previously described[27].

Briefly, all solutions were prepared using ultra-pure RNase-free water and analytical grade reagents. Cells were Harvested using ice cold 1% NP-40 RIP buffer containing 20 mM HEPES-KOH, 5 mM MgCl$_2$, 150 mM KCl, 1 mM DTT, 1% NP-40, 200 U/mL RiboLock, 100 μg/mL Cycloheximide, 1X cOmplete EDTA-free protease Inhibitor. Lysate was centrifuged for 15 min at 15,000 × g. Protein concentration was determined with Bradford protein assay.

For immunoprecipitation, 3 mg of protein were pre-cleared for 1 h and then divided into TDP43-RIP or IgG control tubes, each was incubated overnight at 4 °C with 2 μg of TDP-43 or IgG antibody respectively. Agarose Beads were cleared with RIP buffer at 4 °C for 2 h. TDP43-RIP and IgG samples was incubated with 2 μg antibody. Samples were incubated with beads for 2 h at 4 °C, and then centrifuges at 1,000 × g for 1 min. Beads in pellet were washed 3 times with wash buffer containing 20 mM HEPES-KOH, 5 mM MgCl$_2$, 350 mM KCl, 1 mM DTT, 0.1% NP-40, 200 U/mL RiboLock, 100 μg/mL Cycloheximide and cOmplete EDTA-free protease inhibitor cocktail. RNA from IP fraction was extracted using miRNeasy micro kit (Qiagen). ΔcT was calculated between the cT of each IP sample and its input sample.

**Mitochondria membrane potential measurement**. Mitochondrial membrane potential was assessed with Tetramethylrhodamine Ethyl, Ester (TMRE; Thermo Fisher) dye. Cultures were incubated with 20 nM TMRE for 30 min in CO$_2$ incubator, then washed 3 times with culture medium. Images of TMRE labeled axons in the distal compartment of MFCs were acquired before and after cultures were treated with Anisomycin (40 μM) or NaAsO$_2$ (250 μM), Cycloheximide

(150 μM) in X60 magnification. The volume of medium was kept higher in the proximal compartment to prevent the flow of treatment and ensure cell bodies remain unaffected. The intensity of TMRE fluorescence in axonal mitochondria was measured using FIJI, and the fraction of change post/pretreatment was calculated for each image field.

**Mito-KillerRed (MKR) experiments**. MKR construct was a kind gift from Prof. Thomas Schwartz (Boston Children's Hospital).

MKR experiments were performed by co-culturing WT MNs and muscles in MFCs. Immediately after their plating, MNs were infected with lentiviral particles containing the MKR transfer plasmid. WT ChAT MNs were used as a negative control in co-cultures without MKR expression. After 12 days in co-culture, upon NMJ formation, co-cultures were labeled with Oregon-Green BAPTA (OGB; life technologies) for 40 min. Axons expressing MKR or ChAT were tracked until their contact points with muscles in the distal compartment. An ROI was then marked around the axons that overlap with the muscles, which was later frapped with a 560 nm laser (100 repeats of 200μS). High-speed image sequences of calcium transients were acquired before and after 560 nm laser irradiation, and the percent of change in contraction rate post/pre was calculated for each muscle.

**Lentivirus production and infection**. Lentivirus particles were used to infect MNs with the MKR gene. We used second generation packaging system. The helper pVSVG and pGag-Pol were gifts from Prof. Eran Bacharach (Tel-Aviv University). For lentiviral production, HEK293-T cells (CRL-3216, ATCC) grown on a 60-mm dish. Once 70 to 80% confluence was achieved, cells were transfected with 10 μg of transfer plasmid, 7.5 μg of pGag-Pol, and 2.5 μg of pVSVG. Plasmids were placed in a calcium-phosphate transfection mix (25 mM Hepes, 5 mM KCl, 140 mM NaCl, and 0.75 mM Na$_2$PO$_4$ with 125 mM CaCl$_2$) immediately before their addition to cells, in a volume of 0.5 ml per plate. Culture supernatants were harvested 2 days after transfection and concentrated ×10 using PEG Virus Precipitation kit (Abcam). Final pellets were resuspended in Neurobasal media, aliquoted, and kept in −80 °C until use. For transduction of MNs, 2 μL of concentrated lentiviral suspension was used per MFC containing 150,000 MNs. Lentiviral vectors were added 1 to 2 h after plating MNs and were washed out three times in CNB medium 24 h later.

**Protein synthesis inhibition functional analysis**. Analysis of axon degeneration following protein synthesis inhibition was performed by culturing either WT MNs from HB9::GFP embryos, or ΔNLS and control MNs from TDP43ΔNLS embryos in the proximal compartment of MFC. Once axons have extensively crossed to the distal compartment, or after 10 days (for TDP43ΔNLS cultures), protein synthesis inhibitors were added to the distal (axonal) compartment while maintaining a higher volume of medium in the proximal compartment to prevent exposure of the cell-bodies to inhibitors. Puromycin (100 μg/mL) or Anisomycin (40 μM; only for HB9::GFP MN) were applied exclusively to axons in CNB medium. Images of axons were acquired at low magnification before, and after 16 (TDP43ΔNLS) and 24 h to monitor the extent of axon degeneration.

Analysis of NMJ function following protein synthesis inhibition with puromycin was performed by co-culturing WT MNs with primary muscles transfected with empty PQCXIP-mCherry vectors expressing PAC gene for puromycin resistance. After 12 days in co-culture, once cultures matured and NMJ were formed, Puromycin (100 μg/mL) was added exclusively to the distal (NMJ) compartment for 16 h. The proximal compartment was kept with higher volume of medium for allowing puromycin to act only locally within the NMJ compartment. After 16 h, cultures were labeled with OGB, and the calcium activity of axons and muscles was recorded. Analysis of the percent of muscles with calcium transients in co-culture, was performed on muscles with at least one overlapping axon. Only muscles that expressed mCherry (as a reporter for the expression of PAC) were used for this analysis.

**Co-culture calcium imaging**. OGB lyophilized stock (Life technologies) was resuspended with 20% (w/v) Pluronic acid for a stock concentration of 3 mM Stock was diluted 1:1,000 in the appropriate medium. OGB was incubated with cultures for 40 min in 37 °C, 5% CO$_2$ incubator, and then washed 3 times with culture medium prior imaging. Calcium transients in axons and muscles were recorded in a spinning disk confocal microscope equipped with an EMCCD camera with X40 oil objective using 488 nm laser. Image sequences of 1,000 frames were acquired at frame rate of 25 FPS. Image analysis was performed using the "Time Series Analyzer V3" plugin for FIJI. Briefly, the mean OGB values for a Region of Interest (ROI) were plotted over the complete movie length. This assisted us to determine whether or not a certain muscle was active, and whether the activity was paired with neuronal firing. For figure labeling, axon endings on muscles, which also had high basal OGB signal were considered as NMJs.

**Statistical analysis**. Statistical parameters and test used are noted in figure legends. All experiments included at least 3 biological repeats. Images and micrographs are representative of all experimental repeats. Statistical significance was determined using student's-t-test or mann-whitney test when comparing between two groups, and multiple comparisons ANOVA test when comparing

more than two groups. Multiple comparisons were corrected using Holm-Sidak correction. Threshold for determining statistical significance was $P < 0.05$. All statistical analysis was performed with Graph-pad Prism 7.

**Reporting summary**. Further information on research design is available in the Nature Research Reporting Summary linked to this article.

## Data availability

The proteomics data (Fig. 4a–c) have been deposited in the ProteomeXchange Consortium via the PRIDE partner repository database under accession code PXD021876. Acquired proteomics data was correlated to the mouse FASTA databased using MaxQuant (v. 1.5.3.8) and its implemented Andromeda search engine. All the data generated in this study are provided in the Supplementary Information/Source Data file Source data are provided with this paper.

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

## Acknowledgements

We thank M. Fainzilber and E. Hornstein for helpful comments on the manuscript. The work in E.P lab was supported by the ISF (Israel Science foundation), Ministry of Science and Technology State of Israel, European Research Council (ERC), Israel Ministry of health JPND program, Human Frontiers Science Program (HFSP), grants, 735/19, and 0601166782, 309377, 3-17160, RGP0026/2020-102 respectively, and Radala Foundation for ALS Research. M.E.W lab was supported (in part) by the Intramural Research Program of the National Institutes of Neurological Disorders and Stroke, NIH.

## Author contributions

T.A. and Ar.I. performed cell culture, biochemical and staining experiments, imaging and analysis. Am.I. performed NMJ and sciatic nerve staining, RIP and Western blot. D.P. and M.K. lead the proteomics analysis. T.G.P. was responsible for iPSC cell culture and for animal handling. L.F. Ar.I and N.S. executed RT-PCR analysis. G.A., N.S. and A.D. provided human patient samples. F.R. aided with FISH experiments. R.D., K.T and M.E.W provided iPSC lines. T.A., Ar.I. and E.P. designed the study and wrote the manuscript.

## Competing interests

The authors declare no competing interests.
