## [Peer review file · Nature Communications]

REVIEWER COMMENTS

Reviewer #1 (Remarks to the Author):

This is a very interesting and potentially significant study by Altman et al. to implicate a role of axonal TDP-43 containing RNA granules to inhibit local protein synthesis. These dynamics are impaired in axons of human ALS iPSC derived motor neurons and mouse models with increased levels of cytoplasmic and axonal TDP43 that becomes hyperphosphorylated. In a mouse mutant lacking the NLS for TDP43 with abnormally high levels of axonal TDP43 granules, the inhibition of protein synthesis disrupts NMJ architecture in vivo. Further analysis indicates a selective effect on the level of mitochondrial proteins in axons and at the NMJ. The mouse mutant showed impaired local protein synthesis in association with mitochondria. This study has the potential to provide important conceptual advances in our understanding of the physiologic and pathologic regulation of RNA granule dynamics in distal MN axons to regulate NMJ structure and function. However, there are several technical issues that detract from the story and need to be solidified to support the proposed model. There are also concerns with how this proposed model on gain of cytoplasmic function of TDP43 relates to the well-characterized neuropathology of loss of nuclear TDP43 due to large perinuclear aggregates in ALS.

Specific comments –

The important conceptual advance is that the dynamics of TDP43 granules and local protein synthesis is fundamental to maintain active NMJs. This was revealed in the mouse mutant with abnormally elevated cytoplasmic TDP43. The authors should be careful to soften claims that this is the mechanism in C9orfALS which has multiple other potential pathologic mechanisms e.g. dipeptide repeat proteins, nuclear RNA foci, perinuclear aggregates of TDP43. The mouse model used in this study lack these neuropathologic features that also occur in human ALS postmortem brain. Thus, the statements that “these findings support an axonal gain of function of TDP-43 in ALS” are not warranted. Nonetheless, this study has important implications for contributory mechanisms in ALS that warrant further study. If the authors want to make stronger claims, then new experiments are needed in other ALS mouse models bearing C9orf repeat expansions.

A hallmark neuropathologic feature of human ALS postmortem brain are cytoplasmic aggregates of TDP43 in the cell body that are hyperphosphorylated. The authors use iPSC derived motor neurons from C9orfALS and the mouse mutant lacking TDP43 NLS. No images of cell bodies are shown. Are TDP43 aggregates present in cell bodies? If not, this further warrants the need to downplay claims of ALS relevant neuropathology.

Fig. 1A-C. In the experiments using iPSC derived neurons, need to clarify how many ALS patient lines were used. What does each data point refer to? One patient cell line, or different cells from one line?

Fig. 1E why do control iPSC MN have moderate levels of p-TDP43 in axons? These levels should be negligible.

Fig. 1N. Need to show p-TDP43 levels.

“We found a 2-fold increase in axonal TDP-43 levels upon dox-retraction from MNs (Fig.1N lower panel, and 1O).” – need to show also images of cell body and proximal axon

Fig. 2G shows similar levels of p-TDP43 in control and ALS IM nerves. This image conflicts with 1A.

Fig 2 – need to show colocalization in cell bodies of TDP-43 and G3BP.

Fig 3C needs to show effects C9 peptide on G3BP granules simultaneous with protein synthesis labeling using puromycin.

"Importantly, this decrease in translation was completely abolished upon application of Tat171 fused peptide corresponding to residues 190-208 of G3BP120 exclusively to the axonal 172 compartment (Fig.3c-d)." –need to show effects of peptide on axonal granules

Fig 4 proteome analysis of TDP43 mutant mice suggest reduced axonal mitoproteome. Need some validation in this mouse model that mRNAs are localized, reduced axonal levels in mutant neurons, using cultured neurons as used in Fig 1.

Soften the claim that axonal mRNAs encode mitochondrial proteins, unless use Puro-PLA or other method to document their local translation.

Reviewer #2 (Remarks to the Author):

The authors report the interesting findings that cytoplasmic and axonal accumulation of TDP43 attenuates translation of nuclear-encoded mitochondrial protein mRNAs in primary motor neurons and human iPSC-derived motor neurons based on puromycinylation and puro-PLA studies. Phosphorylated TDP43 similarly accumulates in axons of C9orf72 mutant human neurons, with axonal TDP43 signals colocalizing G3BP1 in both TDP43ΔNLS motor neurons and C9orf72 mutant human neurons. Together, I think these could represent an advance appropriate for consideration by Nature Communications. Also the point that TDP43 accumulations can be removed from axons by attenuating expression (at least for the TDP43ΔNLS) is very intriguing. But I think there are a number of weaknesses that the authors would need to address before further consideration. Some of these are clarifications in the text, but the issue on linking mitochondrial to protein synthesis will need additional experiments.

At several points in the manuscript, the authors indicate that TDP43 is phosphorylated locally in the axons. This is first mentioned in the results on lines 133-134, with phrase 'where it is phosphorylated' and then at a few other sites. However, the present not data to indicate an axon intrinsic phosphorylation event nor whether the phosphorylation precedes accumulation.

The authors suggest an increase in axonal G3BP1 levels for Fig S3. It is not clear if they are imaging overall G3BP1 vs. G3BP1 granules. They need to be specific here and the only way to know of an increase in overall G3BP1 levels would be immunoblotting since the imaging methods end up with the granules saturated to visualize soluble G3BP1.

The use of puromycin resistant muscle co-cultured with neurons is a clever approach that allowed the authors to selectively visualize pre-synaptic protein synthesis. The conclusions would be strengthened by supplemental data showing that none of the OPP incorporation seen is post-synaptic (i.e., the myotubes are in fact puromycin resistant for this assay).

The authors use anisomycin and arsenic treatments to conclude that blocking axonal protein synthesis and increasing stress granule formation decreases mitochondrial function. On one hand, the sum of their data support this conclusion – on the other hand, anisomycin has off target effects on stress kinases and the stress caused by NaASO₂ could affect mitochondrial function by the same mechanisms that it triggers stress granule formation.

Proteomics data show decrease in nuclear encoded mitochondrial proteins in sciatic nerve axoplasm of the TDP43ΔNLS mice and RTqPCR analyses suggest that this is from decreased axonal levels of the mRNAs encoding these proteins. This suggests decreased transport into axons or survival in 'ALS'

axons for these mRNAs. However, FISH and RIP studies indicate a strikingly increased colocalization of candidate mRNAs with axonal TDP43 RNPs. This suggests that the mRNAs are sequestered by the TDP43 granules. Both mechanisms could effectively explain the proteomics, but I think the authors need to reconcile which is the prevailing mechanism. They are certainly not mutually exclusive, but as written these are conflicting mechanisms. Giving the reader some idea of the overall Cox4i2 mRNA levels seen by FISH in these axons and the input levels from the RIP data could resolve this.

The experiments on restoring mitochondrial function and gaining motor reinnervation on halting TDP43 Δ NLS expression would gain more translational relevance by testing the G3BP1 190-208 peptide in this setting. Specifically, if this restores protein synthesis in the ALS axons in absence of muscle, will it also restore mitochondrial function and bring motor reinnervation? EM images in the refenced Liao et al. (2019) paper on transport functions for G3BP1 show axonal G3BP1 granules in very close proximity to mitochondria.

Minor Points:

The authors use the non-standard abbreviation 'LPS' for localized protein synthesis. I suggest that they be given some leeway for word counts and spell this out since LPS has other meanings to many neurobiologists.

Line 114 - phosphorylates should be phosphorylated

Line 116-119 - it would be useful to clarify whether this is sporadic ALS or familial, and if latter which familial?

References – need attention. Volume numbers for Science are odd.

Reviewer #3 (Remarks to the Author):

The manuscript "Axonal TDP-43 Drives NMJ Disruption through Inhibition of Local Protein Synthesis" reports findings that support an axonal gain of function of TDP-43 in ALS, which may potentially be of interest for therapeutic development. The authors report TDP-43 accumulation in intra-muscular nerves from ALS patients and in patient-derived iPSC motor neuron axons, as well as in MNs and NMJs of the dox-repressable human TDP-43 Δ NLS mouse model. Axonal TDP-43 was found to promote G3BP1-positive RNA granule assembly, inhibiting local protein synthesis, especially of nuclear-encoded mitochondrial proteins, in distal axons and NMJs. Shutting down hTDP-43 Δ NLS expression or dissociation of G3BP1 granules restored local translation and rescued TDP-43-derived toxicity in both axons and the NMJ.

This manuscript addresses a topic of general interest that has not been studied much until now. Previous work from several labs has shown that TDP-43 is present in RNA granules in axons of motor neurons and increased TDP-43 protein levels repress axon outgrowth, and different features for mutant TDP-43 granules and a function in RNA localization have been shown. A role in local axonal RNA metabolism has been discussed by several authors, but the hypothetical functional connection to TDP-43 proteinopathies is not well established at this point. For this reason it seems a timely and important study.

However, while the findings are very interesting, there are major issues with essential controls, alternative explanations, and over-interpretation of data that should be addressed to better support the claims in this manuscript, as described below:

1) The manuscript shows a massive increase in (p)TDP-43, G3BP, and RNA in "ALS-axons" (whether animal model or patient derived). It also shows a decrease of certain mitochondrial proteins and

upregulation of ribosomal proteins. This could indicate a very general increase of RNP localization into axons but the manuscript is focused on a very specific interpretation where TDP-43 causes an inhibition of local translation. This is interesting, plausible, and very much possible, but needs to be demonstrated. Is there also an increase in other RNPs, such as e.g. FMRP and Staufen-containing ones?

2) Is there a decrease of mitochondria in these axons? Mitochondria are stained in 4H but not quantified in axons or NMJs, which would be important. This is also relevant since the decreased abundance of axonal mitochondrial proteins in ALS models may be due to a decrease in axonal mitochondria not just related to local translation. It should also be noted that Syto dyes are known to also label mitochondria, which is normally addressed by co-staining with e.g. mitotracker + SytoRNA.

3) Another important question is whether the reduction of translation is axon-specific or also found in the soma? Does TDP-43 mislocalization cause a general reduction in de novo protein synthesis throughout MNs?

4) The manuscript often refers to axonal TDP-43 "aggregates", "pathological RNP granules", or "condensates", but how these are defined is unclear. In the case of aggregates, this can only be addressed by determining e.g. detergent solubility as their defining biochemical feature. How are "aggregates" shown in microscopy different from TDP-43-containing RNP granules?

5) Short of an actual specific depletion of TDP-43 from axons (which is feasible but difficult) it is not clear whether effects are only due to axonal TDP-43 toxicity or nuclear depletion causing e.g. splicing defects. Shutting down hTDP-43^{delNLS} expression leads also to at least partial recovery of nuclear mTDP-43 as shown in Suppl. Fig. 7. This possibility should at least be discussed.

6) Colocalization of transcripts with OPP puncta is not a reliable measure of its local translation, due to diffusion of labeled peptides (Enam et al., eLife 2020 "Puromycin reactivity does not accurately localize translation at the subcellular level"). A better approach would be proximity ligation assay (PLA) - puromycylation to detect newly-synthesized molecules of a specific target protein by co-localization of a protein-specific antibody with an antibody recognizing puromycin.

7) The co-localization of OPP puncta with mitochondria in Fig. 4H may be due to translation in mitochondria. OPP should also label mt-DNA encoded nascent peptides, but in a cycloheximide resistant manner.

8) The manuscript Wang et al., J Neurochem. 2008 May;105(3):797-806. ("TDP-43, the signature protein of FTL-DU, is a neuronal activity-responsive factor") seems relevant and should be cited. This study found that TDP-43 is localized in the dendritic processing (P) body and it behaves as a translational repressor in an in vitro assay and repetitive stimuli by KCl enhanced the colocalization of TDP-43 granules with RNA-binding proteins known to regulate mRNA transport and local translation in neurons. While this study is not focused on axons, it first discusses a potential role for TDP-43 in repression of local translation in neurites.

9) RT-qPCR analysis of the TDP-43 bound mRNAs show a more than 6-fold increase in the binding of both Cox4i1 and ATP5A1 to TDP-43 (Fig.4f-g) in the diseased line. What are the relative transcript levels of Cox4i1 and ATP5A1 transcript in ALS and control iPS-MNs?

10) Supplementary Figure 5 indicated HB9 staining in MN axons. This seems odd, considering that HB9 is a nuclear transcription factor used as a marker for MN nuclei, not axons. (This is different for HB9-driven GFP expression, of course.)

11) Does axonal TDP-43 lead to mitochondrial dysfunction in axons? The manuscript provides no evidence for "synaptic mitochondria toxicity" as shown in the model.

12) Suppl. Fig. 1B. The finding of TDP-43 loss from the nucleus and mislocalization in ALS patient-derived iPSCs into the cytoplasm is surprising, since a slight increase in soluble cytoplasmic TDP-43 is more typically observed in the field. Is this 1 shown cell representative? There is no quantification for this phenomenon shown. The high abundance of TDP-43 outside the nucleus even in control cells raises concerns about the specificity of the staining.

Minor points:

Capitalized letters (A,B,C...) are used in figures but non-capitalized in figure legend (a,b,c...).

Fig. 1N/O only show quantification of axonal TDP-43. How much are TDP-43 levels increased in the soma?

Presumably tTDP43 and tERK in Fig. 1 stands for "total" proteins but is not explained in the legend or text.

"b, d) Quantification of TDP-43(c) or pTDP-43(e) signal within" should be "TDP-43(b) or pTDP-43(d)"
Size bars in H and J are indicated as 10um but seem at different magnifications.

Figure 7 is less than clear and even with its legend hard to understand.

Point by point answer to the reviewer comments

Response to Reviewer #1

This is a very interesting and potentially significant study by Altman et al. to implicate a role of axonal TDP-43 containing RNA granules to inhibit local protein synthesis. These dynamics are impaired in axons of human ALS iPSC derived motor neurons and mouse models with increased levels of cytoplasmic and axonal TDP43 that becomes hyperphosphorylated. In a mouse mutant lacking the NLS for TDP43 with abnormally high levels of axonal TDP43 granules, the inhibition of protein synthesis disrupts NMJ architecture in vivo. Further analysis indicates a selective effect on the level of mitochondrial proteins in axons and at the NMJ. The mouse mutant showed impaired local protein synthesis in association with mitochondria. This study has the potential to provide important conceptual advances in our understanding of the physiologic and pathologic regulation of RNA granule dynamics in distal MN axons to regulate NMJ structure and function. However, there are several technical issues that detract from the story and need to be solidified to support the proposed model. There are also concerns with how this proposed model on gain of cytoplasmic function of TDP43 relates to the well-characterized neuropathology of loss of nuclear TDP43 due to large perinuclear aggregates in ALS.

Response – We thank the reviewer for her/his keen appreciation of our study and hope she/he will find our revisions satisfactory to relieve her/his concerns.

Specific comments –

The important conceptual advance is that the dynamics of TDP43 granules and local protein synthesis is fundamental to maintain active NMJs. This was revealed in the mouse mutant with abnormally elevated cytoplasmic TDP43. The authors should be careful to soften claims that this is the mechanism in C9orfALS which has multiple other potential pathologic mechanisms e.g. dipeptide repeat proteins, nuclear RNA foci, perinuclear aggregates of TDP43. The mouse model used in this study lack these neuropathologic features that also occur in human ALS postmortem brain. Thus, the statements that “these findings support an axonal gain of function of TDP-43 in ALS” are not warranted. Nonetheless, this study has important implications for contributory mechanisms in ALS that warrant further study. If the authors want to make stronger claims, then new experiments are needed in other ALS mouse models bearing C9orf repeat expansions.

Response – We thank the reviewer for these insights. In this work, we show that axonal TDP-43 condensates are apparent in several disease models:

- 1. Sporadic ALS intra-muscular nerves from patient muscle biopsies.*
- 2. C9ORF72 iPSC-derived motor neurons.*
- 3. TDP43 Δ NLS mice model in vitro and in vivo.*

As we show, TDP-43 axonal condensates reduce local translation by sequestering mRNA of nuclear encoded mitochondria proteins, leading to mitochondria alterations, NMJ dysfunction and axon degeneration. In our revised manuscript, we further provide proof that this process can be reversed not only by reducing axonal TDP-43 levels (Fig.7, Fig.8), but also by dissolving axonal G3BP1-TDP-43 condensates (Fig.3c-f, Fig.S7), which subsequently restores axonal translation

(Fig.3g-h), increases functionality of NMJ mitochondria (Fig.5i-j) and recovers NMJ activity deficits (Fig.8f). Specifically, for C9ORF72, we show the existence of axonal G3BP1-TDP-43 condensates (Fig.1e-f, Fig.2d-f), their binding to mRNA (Fig.4k-o), and the ability of G3BP1 inhibiting peptides to dissolve them (Fig.3c-f). Importantly, dissociation of TDP-43 condensates was achieved using local administration of G3BP-1 inhibiting peptides only to the axonal/NMJ side of the MFC, without affecting motor neuron cell body. Therefore, our new data support an axonal gain of function mechanism for TDP-43. This does not contradict the well-established loss of nuclear function of TDP-43, but rather most likely complements it. We stress this point in the main text (introduction line 8), and also note that other potential pathologic mechanisms may provide additional complementation.

A hallmark neuropathologic feature of human ALS postmortem brain are cytoplasmic aggregates of TDP43 in the cell body that are hyperphosphorylated. The authors use iPSC derived motor neurons from C9orfALS and the mouse mutant lacking TDP43 NLS. No images of cell bodies are shown. Are TDP43 aggregates present in cell bodies? If not, this further warrants the need to downplay claims of ALS relevant neuropathology.

Response – We thank the reviewer for this important comment. To show that our cellular models recapitulate the basic features of TDP-43 pathology as seen in ALS patients, we added a new experiment in the revised version in which we stained cell bodies with antibodies for phosphorylated TDP-43 and G3BP1. The results show both cytoplasmic mislocalization of TDP-43 in its regular (Fig.S1b) and phosphorylated form (Fig. S4, attached below). We further performed careful 3D analyses to quantify the extent of colocalization between phosphorylated-TDP-43 and G3BP1 in cell-bodies and proximal axons of both C9ORF72 iPSC-motor neurons and TDP43 Δ NLS motor neurons (Fig.S3a-f, Fig S4, attached below). Thus, we provide evidence that the ALS models we use exhibit the presence of phosphorylated TDP-43 condensates, as found in ALS post-mortem tissues, in all neuronal compartments (soma, proximal and distal axon) in several cellular modalities.

Reviewer 1, Fig. 1 (Cropped from Fig. S3 in revised manuscript with relevant experiment) – a-d) Representative images (a) and quantitative analysis of pTDP-43-G3BP1 colocalization (b), G3BP1 intensity (c), and pTDP-43 intensity (d) in cell bodies of TDP43 Δ NLS motor neurons. e-f) Representative images and quantitative analysis of pTDP-43-G3BP1 colocalization in proximal axons of TDP43 Δ NLS motor neurons.

Reviewer 1, Fig. 2 (Cropped from Fig. S4 in revised manuscript with relevant experiment) – a-d) Representative images (a) and quantitative analysis of pTDP-43-G3BP1 colocalization (b), G3BP1 intensity (c), and pTDP-43 intensity (d) in cell bodies of C9ORF72 iPSC-motor neurons. e-f) Representative images and quantitative analysis of pTDP-43-G3BP1 colocalization in proximal axons of C9ORF72 iPSC-motor neurons.

Fig. 1A-C. In the experiments using iPSC derived neurons, need to clarify how many ALS patient lines were used. What does each data point refer to? One patient cell line, or different cells from one line?

Response – The samples analyzed in figures 1A-C are of adult ALS patient muscle biopsy samples, and not iPSC lines. Each data point refers to the average TDP-43 or phosphorylated TDP-43 in the intra-muscular nerve of a single patient, whose details are found in the methods section. In other instances, we use C9ORF72 motor neurons derived from one ALS patient iPSCs and compare our findings with the corrected-isogenic control line derived from the same patient.

Fig. 1E why do control iPSC MN have moderate levels of p-TDP43 in axons? These levels should be negligible.

Response – We improved picture quality, as seen in the new updated figure (Fig. 1E, attached below). Under these sensitive imaging conditions, the new image

clearly indicates the difference between control and mutant iPS-motor neurons. This result has been replicated in several other experiments throughout the revised manuscript (Fig 2d, Fig 4m, Fig. S4e), providing firm evidence that it represents true differences between these lines.

Reviewer 1, Fig 3. (Cropped from Fig. 1 of the revised manuscript). Corrected IF images indicating that, although still present to some extent, the levels of pTDP-43 in axon of the control iPS-motor neurons are significantly lower compared to its levels in C9ORF72 iPS-motor neurons axons.

Fig. 1N. Need to show p-TDP43 levels.

Response – We now added new figure Fig.1n-p and Fig.S2g (attached below) where we show a significant 2-fold increase in phosphorylated-TDP43 in western blot of axonal purified protein lysates.

Reviewer 1, Fig. 4 – Western blot and analysis of phosphorylated-TDP-43 levels in protein lysates acquired from distal axons of TDP43ΔNLS motor neurons cultured in radial microfluidic chambers we developed that separate the axonal compartments from the soma (Fig.S2c-d).

“We found a 2-fold increase in axonal TDP-43 levels upon dox-retraction from MNs (Fig.1N lower panel, and 1O).” – need to show also images of cell body and proximal axon.

Response – We now have new images and quantification of phosphorylated TDP-43 levels in soma and proximal axons in both TDP43ΔNLS and C9ORF72-iPS models (Fig.S3, Fig.S4, Reviewer 1 Figures 1&2 in this letter).

Fig. 2G shows similar levels of p-TDP43 in control and ALS IM nerves. This image conflicts with 1A.

Response – We improved image quality in the revised version to show better representation of the data.

Fig 2 – need to show colocalization in cell bodies of TDP-43 and G3BP.

Response - We now have 2 additional supplementary figures, Fig.S3-S4 (Reviewer 1 Figures 1&2 above), that demonstrate and quantify the increase in colocalization of phosphorylated TDP-43 and G3BP1 in both TDP43ΔNLS and C9ORF72-iPS models.

Fig 3C needs to show effects C9 peptide on G3BP granules simultaneous with protein synthesis labeling using puromycin.

“Importantly, this decrease in translation was completely abolished upon application of Tat fused peptide corresponding to residues 190-208 of G3BP120 exclusively to the axonal compartment (Fig.3c-d).” –need to show effects of peptide on axonal granules.

Response - To better characterize the effect of the G3BP1 peptide on axonal condensate formation and axonal protein synthesis, we performed additional experiments. We demonstrate that application of G3BP1(190-208) peptide exclusively to distal axons dissolves G3BP1 axonal granules and reduces the pTDP43-G3BP1 particle size and density, resulting in a simultaneous increase in axonal protein synthesis (Fig.3c-h, Fig.S7). Attached below is a combined version of the experimental results prepared for this response.

Reviewer 1, Fig. 5 (Combined version of experimental results gathered from Fig. 3 and Fig. S7 in the revised manuscript) - Representative images and quantification of G3BP1, pTDP43, and their co-localization particle densities and sizes in correlation with the amount of axonal translation in G3BP1 peptide-treated and untreated C9ORF72 and control iPS-motor neurons. G3BP1 peptide was applied exclusively to axons in the distal compartment of microfluidic chambers.

Fig 4 proteome analysis of TDP43 mutant mice suggest reduced axonal mitoproteome. Need some validation in this mouse model that mRNAs are localized, reduced axonal levels in mutant neurons, using cultured neurons as used in Fig 1.

Response – We performed additional validations of the proteome analysis to show reduction in axonal levels of nuclear encoded mitochondrial proteins, specifically Cox4i and ATP5A1. The validations demonstrate a decline in protein levels in vivo in TDP43ΔNLS sciatic nerves (Fig.4d-g, attached below) and in vitro in cultured TDP43ΔNLS neurons (Fig.4h, attached below). Furthermore, we conducted RT-qPCR for Cox4i1 and ATP5A1 that reveal slight increases in their axonal mRNA levels in the TDP43ΔNLS model both in vivo (sciatic nerve axoplasm) and in vitro (cultured neuronal axons); (Fig.4i-j, attached below). This data is supported by new TDP-43-RNA immunoprecipitation from axons of C9ORF72 and control iPSC-motor neurons (Fig. 4k-i, attached below), and by combined smFISH-IF experiments for Cox4i1 mRNA together with G3BP1 which demonstrated higher Cox4i1 mRNA levels in axons which are sequestered by pTDP-43-RNP condensates (Fig 4.m-o, attached below) that indicated TDP-43 sequesters these transcripts.

Reviewer 1, Fig 6. (cropped from Fig. 4 in the revised manuscript) - In-vivo and in-vitro validations for proteome analysis demonstrating reduction in Cox4i and ATP5A1 protein levels in the TDP43ΔNLS model. d-g) Immunofluorescence images and quantification of Cox4i and ATP5A1 levels within ChAT-positive axons in sciatic nerves. h) Western blot analysis of Cox4i levels in protein lysates of isolated TDP43ΔNLS motor neuron axons cultured in radial microfluidic chambers.

Reviewer 1, Fig. 7 (Cropped from Fig. 4 in the revised manuscript) – i) RT-qPCR for ATP5A1, Cox4i1 and Ndufa4 from sciatic axoplasm pure axonal fraction of TDP43ΔNLS and control motor neuron cultures grown in radial microfluidic chamber. j) RT-qPCR for ATP5A1, Cox4i1 and Ndufa4 from pure axonal fraction of TDP43ΔNLS and control motor neuron cultures grown

in radial microfluidic chamber. k-i) western blots and RT-qPCR quantification of Cox4i1 and ATP5A1 from TDP-43-RIP performed on either somata, or isolated axons of C9ORF72 iPS-motor neurons and isogenic controls.

Reviewer 1, Fig 8. (Cropped Fig.4m-o in the revised manuscript) – Combined Cox4i1 smFISH and IF staining for G3BP1 and pTDP-43 demonstrate sequestration of Cox4i1 mRNA in TDP-43-RNP condensates within C9ORF72 iPS-motor neurons.

Reviewer 1, Fig 9. (prepared for this response and not included in the revised manuscript) – smFISH quantification of axonal Cox4i1 mRNA levels in C9ORF72 iPS-motor neuron axons.

Soften the claim that axonal mRNAs encode mitochondrial proteins, unless use Puro-PLA or other method to document their local translation.

Response – To validate the local translation of nuclear encoded mitochondrial proteins, specifically Cox4i, we performed streptavidin pull downs of biotin-OPP from TDP43 Δ NLS sciatic nerves (Fig.4p, Fig.S10b-c, attached below). Our new data shows Cox4i pulldown with OPP decrease in the TDP43 Δ NLS model, indicating that its local translation in axons is decreased upon TDP-43 mislocalization to the axon.

Reviewer 1, Fig 10. (Combined figure prepared for this response, composed of Fig. S10b-c and Fig. 4p). a) validations for the sensitivity and specificity of Biotin-OPP streptavidin pull-downs performed in Hek293T cells. b) Total streptavidin-HRP blots of sciatic nerves from control, TDP43 Δ NLS, Anisomycin treated, and untagged control (3 mice per lane) c) Immunodetection of Cox4i following biotin-OPP streptavidin pull downs from sciatic nerves (10 mice per lane) suggests reduced Cox4i axonal local synthesis upon TDP-43 mislocalization.

Response to Reviewer #2

The authors report the interesting findings that cytoplasmic and axonal accumulation of TDP43 attenuates translation of nuclear-encoded mitochondrial protein mRNAs in primary motor neurons and human iPSC-derived motor neurons based on puromycinylation and puro-PLA studies. Phosphorylated TDP43 similarly accumulates in axons of C9orf72 mutant human neurons, with axonal TDP43 signals colocalizing G3BP1 in both TDP43 Δ NLS motor neurons and C9orf72 mutant human neurons. Together, I think these could represent an advance appropriate for consideration by Nature Communications. Also the point that TDP43 accumulations can be removed from axons by attenuating expression (at least for the TDP43 Δ NLS) is very intriguing. But I think there are a number of weaknesses that the authors would need to address before further consideration. Some of these are clarifications in the text, but the issue on linking mitochondrial to protein synthesis will need additional experiments.

Response - We sincerely thank the reviewer for this constructive feedback. We have substantially revised our manuscript accordingly, adding new data and clarifying textual inaccuracies. We hope that this revision addresses the reviewer's concerns to her/his satisfaction. Below is a point-by-point response.

At several points in the manuscript, the authors indicate that TDP43 is phosphorylated locally in the axons. This is first mentioned in the results on lines 133-134, with phrase 'where it is phosphorylated' and then at a few other sites. However, the present not

data to indicate an axon intrinsic phosphorylation event nor whether the phosphorylation precedes accumulation.

Response - We thank the reviewer for highlighting this textual inaccuracy. We did not mean to imply that TDP-43 undergoes phosphorylation in axons, but rather to point out that we detect higher levels of the phosphorylated form of TDP-43 in axons. We have now revised the manuscript to better represent our findings.

The authors suggest an increase in axonal G3BP1 levels for Fig S3. It is not clear if they are imaging overall G3BP1 vs. G3BP1 granules. They need to be specific here and the only way to know of an increase in overall G3BP1 levels would be immunoblotting since the imaging methods end up with the granules saturated to visualize soluble G3BP1.

Response - The previous version of the manuscript contained a quantification of the total fluorescence intensity of G3BP1 within axons of TDP43 Δ NLS motor neurons and C9ORF72 iPS-motor neurons. The images in Figure 2 demonstrate that the G3BP1-granules are not saturated compared to the uncondensed G3BP1, hence the intensity measurements are likely representative of the overall G3BP1 levels in axons. Nonetheless, the reviewer's point is well taken, and we carried out isolation of axonal protein content from cultured TDP43 Δ NLS motor neurons and their controls in radial MFCs and validated the increase in overall G3BP1 levels in axons by western blot analysis (Attached below, and in Fig. S3). We also added an additional quantification of G3BP1 particle size and particle density in C9ORF72 iPS-motor neuron axons and their isogenic controls and demonstrate that both are increased in C9ORF72 axons, supporting overall higher levels of G3BP1 in these axons (Attached below, and in Fig 3 & Fig. S7).

Reviewer 2, Fig. 1 – Western blot and quantification indicating an increase in total G3BP1 levels in TDP43 Δ NLS motor neuron axons compared to control axons.

Reviewer 2, Fig. 2 (Combined version of experimental results gathered from Fig. 3 and Fig. S7 in the revised manuscript) – Representative images and quantification of the G3BP1, pTDP43, and their co-localization particle densities and sizes in correlation with the amount of axonal translation in G3BP1(190-208) peptide-treated and untreated C9ORF72 and control iPS-motor neurons. G3BP1 peptide was applied exclusively to axons in the distal compartment of microfluidic chambers.

The use of puromycin resistant muscle co-cultured with neurons is a clever approach that allowed the authors to selectively visualize pre-synaptic protein synthesis. The conclusions would be strengthened by supplemental data showing that none of the OPP incorporation seen is post-synaptic (i.e., the myotubes are in fact puromycin resistant for this assay).

Response - We thank the reviewer for this comment.

Figure S8 (attached below) demonstrates the resistance of puromycin transfected muscles to puromycin labeling. We also prepared an additional figure to specifically address this reviewer concerns regarding distinction between pre and post synaptic OPP signal in resistant muscle (see below Reviewer 2 Fig. 4).

Reviewer 2, Fig. 3 – a-b) Representative images and quantification of the puromycin intensity in control muscles compared with muscles expressing Puromycin N-acetyltransferase (PAC) carried in an empty backbone vector with mCherry reporter. c)

Representative images demonstrating puromycin resistant muscles remain intact following 24-hours of high-dose puromycin treatment as compared with untreated muscles.

Reviewer 2, Fig 4 – Representative images of OPP labeling in resistant versus non-resistant muscle in co-culture demonstrate that no post-synaptic background labeling is visible in resistant muscles.

The authors use anisomycin and arsenic treatments to conclude that blocking axonal protein synthesis and increasing stress granule formation decreases mitochondrial function. On one hand, the sum of their data support this conclusion – on the other hand, anisomycin has off target effects on stress kinases and the stress caused by NaAsO₂ could affect mitochondrial function by the same mechanisms that it triggers stress granule formation.

Response - We have tested the effects of another protein synthesis inhibitor – Cycloheximide, which should not interfere with mtDNA-encoded protein synthesis, over TMRE signal in axonal mitochondria, as was also suggested by reviewer #3. The results of this experiment were added to Fig. 5 of the revised manuscript and are also attached below. Inhibiting axonal protein synthesis in a mitochondria resistant manner, still reduces mitochondrial activity to an extent comparable to anisomycin and NaAsO₂. Additionally, we quantified the colocalization of OPP with mitochondria (MitoTracker) in axons treated with cycloheximide and identified a reduction in the mitochondria-associated newly synthesized proteins (Fig. S11, and below).

Reviewer 2, Fig. 5 (Cropped from fig.5 in the revised manuscript) – Representative images and quantification of TMRE signal in distal axons following application of conventional protein synthesis inhibitors and NaAsO₂.

Reviewer 2, Fig. 6 (Fig S11 in the revised manuscript) – Representative images and quantification of mitochondria colocalization with OPP in Cycloheximide (CHX)-treated versus untreated axons.

Proteomics data show decrease in nuclear encoded mitochondrial proteins in sciatic nerve axoplasm of the TDP43 Δ NLS mice and RTqPCR analyses suggest that this is from decreased axonal levels of the mRNAs encoding these proteins. This suggests decreased transport into axons or survival in 'ALS' axons for these mRNAs. However, FISH and RIP studies indicate a strikingly increased colocalization of candidate mRNAs with axonal TDP43 RNPs. This suggests that the mRNAs are sequestered by the TDP43 granules. Both mechanisms could effectively explain the proteomics, but I think the authors need to reconcile which is the prevailing mechanism. They are certainly not mutually exclusive, but as written these are conflicting mechanisms. Giving the reader some idea of the overall Cox4i2 mRNA levels seen by FISH in these axons and the input levels from the RIP data could resolve this.

Response – In our revised version we provided better explanation for our findings regarding axonal mRNA and TDP-43 condensates, using both textual changes and additional experiments. Our new data suggest that nuclear-encoded mitochondrial mRNAs are sequestered by TDP-43-RNP condensates that results in inhibition of local axonal translation.

As our sciatic axoplasm proteome analysis revealed, we found lower levels of nuclear-encoded mitochondrial proteins in the TDP43 Δ NLS samples. However, the RT-qPCR from same sciatic axoplasm samples demonstrated that their mRNA levels were unchanged and even slightly higher. Further validation by TDP-43-RIP from C9ORF72 iPS-motor neurons, and by smFISH support mechanism of RNA sequestration by TDP-43-RNP condensates. We also strengthen this mechanistic finding by performing RT-qPCR for pure axons of TDP43 Δ NLS and control motor neurons grown and isolated from radial microfluidic chambers (Fig 4j, and attached below). Moreover, we performed TDP-43-RIP from isolated axons of the C9ORF72 iPS-motor neurons and their isogenic controls, which demonstrates sequestration of nuclear encoded mitochondrial mRNAs in axons within TDP-43 RNP condensates (Fig 4k-l and attached below). Lastly, we performed quantitative combined smFISH and IF analysis in C9ORF72 iPS-motor neuron axons to demonstrate the sequestration of Cox4i1 within pTDP-43 and G3BP1 axonal condensates (Fig. 4m-o and attached below).

Reviewer 2, Fig. 7 (Cropped from Fig. 4j-l in the revised manuscript) - Left panel, RT-qPCR for ATP5A1, Cox4i1 and Ndufa4 from pure axonal fraction of TDP43 Δ NLS and control motor neuron cultures grown in radial microfluidic chamber. Right panel, western blots and RT-qPCR quantification of Cox4i1 and ATP5A1 from TDP-43-RIP performed on either somata, or isolated axons of C9ORF72 iPS-motor neurons and isogenic controls.

Reviewer 2, Fig. 8 (cropped from Fig. 4m-o in the revised manuscript) - Representative images and quantification of combined Cox4i1 smFISH and pTDP-43 +G3BP1 immunostaining in axons of C9ORF72 iPS-motor neurons and isogenic control. Quantification shows the percent of Cox4i1 mRNA and pTDP-43 colocalized area out of total Cox4i1 mRNA and the percent of colocalized area between Cox4i1 mRNA, pTDP-43 and G3BP1.

Reviewer 2, Fig 9. (prepared for this response and not included in the revised manuscript) - smFISH quantification of axonal Cox4i1 mRNA levels in C9ORF72 iPS-motor neuron axons.

The experiments on restoring mitochondrial function and gaining motor reinnervation on halting TDP43 Δ NLS expression would gain more translational relevance by testing the G3BP1 190-208 peptide in this setting. Specifically, if this restores protein synthesis in the ALS axons in absence of muscle, will it also restore mitochondrial function and bring motor reinnervation? EM images in the referenced Liao et al. (2019) paper on transport functions for G3BP1 show axonal G3BP1 granules in very close proximity to mitochondria.

Response - We thank the reviewer for this important point. We have seriously addressed it in our revised manuscript and included a new set of data regarding the functional benefit the G3BP1 190-208 peptide delivers to motor neurons. Specifically, as also partially mentioned above (Fig. 1 in the response to this reviewer), we:

- 1. Tested the ability of G3BP1 peptides to dissociate G3BP1, as well as pTDP-43-G3BP1 axonal condensates when applied exclusively to axons (see above, Reviewer 2 Fig. 2).*
- 2. Tested the ability of G3BP1 peptides to recover mitochondrial function in TDP43 Δ NLS co-cultures (Fig. 5i-j and attached below) and*
- 3. Tested the ability of G3BP1 peptides to recover NMJ function in-vitro, as did the TDP-43 recovery paradigm in our previous version (Fig. 8F, attached below).*

Reviewer 2, Fig 10 (cropped from Fig. 5i-j in the revised manuscript) – Representative images and quantification of pre-synaptic TMRE signal intensity in Thy1-mitoDendra-control, Thy1-mitoDendra-TDP43 Δ NLS, and in G3BP1 peptide treated Thy1-mitoDendra-TDP43 Δ NLS axons.

Reviewer 2, Fig. 11 (cropped from Fig. 8f in the revised manuscript) – Quantification of the percent of contracting muscles in control and TDP43ΔNLS co-culture with, or without the exclusive presence of G3BP1 peptides in the axonal/NMJ compartment.

Minor Points:

The authors use the non-standard abbreviation ‘LPS’ for localized protein synthesis. I suggest that they be given some leeway for word counts and spell this out since LPS has other meanings to many neurobiologists.

Response - FIXED – We no longer use this or any other abbreviation for local protein synthesis in the revised manuscript.

Line 114 - phosphorylates should be phosphorylated

Response - FIXED. Thank you.

Line 116-119 - it would be useful to clarify whether this is sporadic ALS or familial, and if latter which familial?

Response - This was further clarified in the current version. The description of all ALS and control patients appears in the method section. All ALS patient tissues were taken from sporadic ALS background.

References – need attention. Volume numbers for Science are odd.

Response - FIXED.

Response to Reviewer #3

The manuscript “Axonal TDP-43 Drives NMJ Disruption through Inhibition of Local Protein Synthesis” reports findings that support an axonal gain of function of TDP-43

in ALS, which may potentially be of interest for therapeutic development. The authors report TDP-43 accumulation in intra-muscular nerves from ALS patients and in patient-derived iPSC motor neuron axons, as well as in MNs and NMJs of the dox-repressible human TDP-43 Δ NLS mouse model. Axonal TDP-43 was found to promote G3BP1-positive RNA granule assembly, inhibiting local protein synthesis, especially of nuclear-encoded mitochondrial proteins, in distal axons and NMJs. Shutting down hTDP-43 Δ NLS expression or dissociation of G3BP1 granules restored local translation and rescued TDP-43-derived toxicity in both axons and the NMJ.

This manuscript addresses a topic of general interest that has not been studied much until now. Previous work from several labs has shown that TDP-43 is present in RNA granules in axons of motor neurons and increased TDP-43 protein levels repress axon outgrowth, and different features for mutant TDP-43 granules and a function in RNA localization have been shown. A role in local axonal RNA metabolism has been discussed by several authors, but the hypothetical functional connection to TDP-43 proteinopathies is not well established at this point. For this reason it seems a timely and important study.

However, while the findings are very interesting, there are major issues with essential controls, alternative explanations, and over-interpretation of data that should be addressed to better support the claims in this manuscript, as described below:

Response – We appreciate the reviewer’s interest and thoughtful input. We have thoroughly revised our manuscript by adding new experiments, and critical controls, as detailed below.

1) The manuscript shows a massive increase in (p)TDP-43, G3BP1, and RNA in “ALS-axons” (whether animal model or patient derived). It also shows a decrease of certain mitochondrial proteins and upregulation of ribosomal proteins. This could indicate a very general increase of RNP localization into axons but the manuscript is focused on a very specific interpretation where TDP-43 causes an inhibition of local translation. This is interesting, plausible, and very much possible, but needs to be demonstrated. Is there also an increase in other RNPs, such as e.g. FMRP and Staufen-containing ones?

Response – We thank the reviewer for this interesting observation. To examine if TDP-43 axonal mislocalization and its colocalization with G3BP1 is a general concept for many RNPs, we performed immunostaining of FMRP or Staufen1 together with G3BP1 in TDP43 Δ NLS motor neuron axons. We did not observe an increase in the colocalization of either FMRP or Staufen1 with G3BP1 in the mutant axons, and we even found a decrease in the colocalized FMRP particles (attached figure below). Therefore, we conclude that TDP-43 mislocalization does not lead to a general RNP axonal propagation, at least in our experimental system.

Reviewer 3, Fig. 1. – Representative images and colocalization analysis of G3BP1 with FMRP (a-b) or with Staufen1 (c-d) in TDP43 Δ NLS axons.

2) Is there a decrease of mitochondria in these axons? Mitochondria are stained in 4H but not quantified in axons or NMJs, which would be important. This is also relevant since the decreased abundance of axonal mitochondrial proteins in ALS models may be due to a decrease in axonal mitochondria not just related to local translation. It should also be noted that Syto dyes are known to also label mitochondria, which is normally addressed by co-staining with e.g. mitotracker + SytoRNA.

Response – We thank the reviewer for this constructive comment. To solidify our hypothesis, we performed several experiments in our revised manuscript:

1. *To validate the proteome findings, we performed immunostaining for Cox4i ,ATP5A1, and immunoblotting of Cox4i protein, and in both methods, we were able to demonstrate a general reduction in protein levels (Fig.4d-h in the revised manuscript, attached below).*
2. *Using RT-qPCR of purified axons, we demonstrate that Cox4i1 general mRNA levels are unchanged or are even slightly increased in TDP43 Δ NLS motor neuron axons (Fig.4j in the revised manuscript, attached below).*
3. *Using combined IF-smFISH method and TDP-43 RNA pull down from purified axons, we show that mRNA of the nuclear-encoded mitochondrial protein, Cox4i1, is associated with TDP-43 in axons and is found within G3BP1 positive axonal condensates (Fig.4k-o in the revised manuscript, attached below).*
4. *To further strengthen this observation, we performed streptavidin pull-downs of biotin-OPP from sciatic nerves of TDP43 Δ NLS and control mice (Fig. 4p in the revised manuscript, attached below). In this assay, we were able to demonstrate substantially lower levels of translated Cox4i in axons.*

Therefore, it is highly likely that there is repression in *Cox4i1* protein translation, that is not directly linked to the reduction in mitochondria numbers.

Reviewer 3, Fig 2. (cropped from Fig.4 in the revised manuscript) - In-vivo and in-vitro validations for proteome analysis demonstrating reduction in *Cox4i* and *ATP5A1* protein levels in the *TDP43ΔNLS* model. d-g) Immunofluorescence images and quantification of *Cox4i* and *ATP5A1* levels within ChAT-positive axons in sciatic nerves. h) Western blot analysis of *Cox4i* levels in protein lysates of isolated *TDP43ΔNLS* motor neuron axons cultured in the radial microfluidic chambers.

Reviewer 3, Fig. 3 (cropped from Fig.4 in the current version of the manuscript) – i-j) RT-qPCR for *ATP5A1*, *Cox4i1* and *Ndufa4* from sciatic axoplasm of *TDP43ΔNLS* and control mice (i) and from pure axonal fraction of *TDP43ΔNLS* and control motor neuron cultures grown in radial MFC (j). k) Western blots and RT-qPCR quantification of *Cox4i1* and *ATP5A1* from TDP-43-RIP performed on either somata, or isolated axons of *C9ORF72* iPS-motor neurons and isogenic controls.

Reviewer 3, Fig. 4 (cropped from Fig. 4m-o in the revised manuscript) - Representative images and quantification of combined Cox4i1 smFISH and pTDP-43 +G3BP1 immunostaining in axons of C9ORF72 iPS-motor neurons and isogenic control. Quantification shows the percent of Cox4i1 mRNA and pTDP-43 colocalized area out of total Cox4i1 mRNA and the percent of colocalized area between Cox4i1 mRNA, pTDP-43 and G3BP1.

Reviewer 3, Fig 5. (prepared for this response and not included in the current manuscript) – smFISH quantification of axonal Cox4i1 mRNA levels in C9ORF72 iPS-motor neuron axons.

Reviewer 3, Fig 6. (Combined figure prepared for this letter composed of Fig. S10b-c and Fig. 4p). a) validations for the sensitivity and specificity of Biotin-OPP streptavidin pull-downs performed in Hek293T cells. b) Total streptavidin-HRP blots of sciatic nerves from control, TDP43 Δ NLS, Anisomycin treated, and untagged control (3 mice per lane) c) Immunodetection of Cox4i following biotin-OPP streptavidin pull downs from sciatic nerves (10 mice per lane) reveal that the local synthesis of Cox4i in axons is greatly reduced upon TDP-43 mislocalization.

However, to further answer this reviewer's concerns we examined whether this reduction in the levels of nuclear-encoded mitochondrial proteins is associated also with direct mitochondria damage that reduce mitochondria numbers. To that end, we bred our TDP43 Δ NLS mice to a mouse line which expresses a fluorescent tag in neuronal mitochondria, the Thy1-MitoDendra mice. Using this combined

mouse model, we show extensive decline in axonal and synaptic mitochondria density (Fig.5 in the revised manuscript, attached below). Thus, we demonstrate that TDP-43 mislocalization leads to direct mitochondria alterations, which are most likely related to the reduction in the local protein synthesis of nuclear encoded mitochondrial proteins. As noted in the discussion section, future experiments will determine the sequence of events to understand if mitochondrial toxicity is preceded by a reduction in axonal protein synthesis, or it might be that a local energy deficiency causes a vicious cycle of disrupted protein synthesis and subsequent damage to axonal and synaptic mitochondria.

Reviewer 3, Fig. 7 (cropped from Fig 5 in the revised manuscript) – e-f) Representative images and quantification of mitochondrial size and morphology in TDP Δ NLS-Thy1-mitoDendra sciatic nerves. g-h) Representative images quantification of percent of mitochondrial volume in NMJs of TDP43 Δ NLS-Thy1-mitoDendra mice.

Following the suggestion that Syto-RNA dye should be used in conjunction with other markers, we performed an additional experiment of co-staining motor neuron axons with mitotracker, Syto-RNA, TDP-43 and G3BP1. In this experiment we were able to demonstrate that despite its affinity to mitochondria, Syto-RNA dye also stains RNA that is mitoTracker negative, and even RNA that is localized to axonal RNP condensates. Therefore, together with our revised version of the combined smFISH for Cox4i mRNA with IF of phosphorylated TDP-43 and G3BP1 (Fig. 4 in the response to this reviewer), the observations regarding the presence of RNA positive RNP-condensates hold and fit our conclusions.

Reviewer 3, Fig 8. (prepared for the response for this reviewer and does not appear in the revised manuscript) – Representative images and quantification of the fraction of TDP-43 RNP-condensates in TDP43 Δ NLS axons compared to the fraction of TDP-43 RNP-condensates which are also colocalized with mitochondria (mitoTracker) in the same axons.

3) Another important question is whether the reduction of translation is axon-specific or also found in the soma? Does TDP-43 mislocalization cause a general reduction in de novo protein synthesis throughout MNs?

Response – To address this important question, we performed new experiments, where we stained cell bodies of TDP43 Δ NLS motor neurons and C9ORF72 iPS-motor neurons and their controls with OPP. In those experiment we show that despite the major decrease in the levels of axonal protein synthesis, somatic protein synthesis was only slightly reduced in C9ORF72 iPS-motor neuron cell bodies, and even mildly increased in TDP43 Δ NLS motor neuron cell bodies (Fig.S6). Thus, we conclude that in our system, the reduction of protein synthesis is mainly axon specific and not necessarily associated with a general reduction in protein synthesis throughout the entire motor neuron.

Reviewer 3, Fig. 9 (cropped from Fig S6 in the revised manuscript) – Representative images and analysis of IF for pTDP-43 and OPP staining in cell bodies of C9ORF72 and control iPS-motor neurons (h-i) and in TDP43 Δ NLS and control motor neuron cell bodies (j-k).

4) The manuscript often refers to axonal TDP-43 “aggregates”, “pathological RNP granules”, or “condensates”, but how these are defined is unclear. In the case of aggregates, this can only be addressed by determining e.g. detergent solubility as their defining biochemical feature. How are “aggregates” shown in microscopy different from TDP-43-containing RNP granules?

Response – We thank the reviewer for this comment. In the text of our revised version of the manuscript we referred to all mislocalized axonal TDP-43 that is G3BP1 positive as TDP-43 RNP-condensates. We believe condensates can be used as a general term to describe RNP granules that are colocalized to G3BP1 protein, a crucial component of stress granule cores. As we also suggest in the discussion of this current manuscript, future work should characterize the biophysical properties and proteins/RNA compositions of those axonal condensates. Unfortunately, this is out of the scope of the current paper.

5) Short of an actual specific depletion of TDP-43 from axons (which is feasible but difficult) it is not clear whether effects are only due to axonal TDP-43 toxicity of nuclear depletion causing e.g. splicing defects. Shutting down hTDP-43deINLS expression leads also to at least partial recovery of nuclear mTDP-43 as shown in Suppl. Fig. 7. This possibility should at least be discussed.

Response – We thank the reviewer for this key insight. As we show in our revised version, TDP-43 axonal condensates reduce local translation by sequestering mRNA of nuclear encoded mitochondria proteins, leading to mitochondria alterations, NMJ dysfunction and axon degeneration. We also display proof that this process can be reversed not only by reducing axonal TDP-43 levels (Fig.7, Fig.8), but also by dissolving axonal G3BP1-TDP-43 condensates (Fig.3c-f, Fig.S6, attached below), which subsequently restores axonal translation (Fig.3g-h), increases functionality of NMJ mitochondria (Fig.5l-m, attached below) and recovers NMJ activity deficits (Fig.8f, attached below). Importantly, the method for dissociation of TDP-43 condensates was achieved using local administration of G3BP-1 inhibiting peptides only to the axonal/NMJ side of the microfluidic chamber, without affecting the motor neuron cell body. Therefore, we believe our new data provides more confidence in a TDP-43 axonal gain of function. This is not in contradiction to the well-established TDP-43 loss of nuclear function, as we mention in the main text (introduction line 8), but as an additional complementary mechanism. The regain of the important roles of nuclear TDP-43 probably convey additive beneficial effects to the recovery model we present, but we believe our new data regarding localized dissociation of axonal condensates and its positive results strengthen the hypothesis for the existence of an additional, axonal gain of function role for TDP-43 in ALS. Furthermore, C9ORF72 iPS-motor neurons do not present total TDP43 nuclear depletion, and TDP43 can be found also at the nucleus (e.g. Fig.S4 and Fig.S6). This can further suggest an axonal TDP43 gain of function in addition to TDP43 nuclear loss of function.

Reviewer 3, Fig. 10 (Combined version of experimental results cropped from Fig. 3 and Fig. S7 in the revised manuscript)– Representative images and quantification of the G3BP1, pTDP43, and their co-localization particle densities and sizes in correlation with the amount of axonal translation in G3BP1(190-208) peptide-treated and untreated C9ORF72 and control iPS-motor neurons. G3BP1 peptide was applied exclusively to axons in the distal compartment of microfluidic chambers.

Reviewer 3, Fig 11 (Cropped from Fig.5 in the revised manuscript) – Representative images and quantification of pre-synaptic TMRE signal intensity in Thy1-mitoDendra-control, Thy1-mitoDendra-TDP43 Δ NLS, and in G3BP1 peptide treated Thy1-mitoDendra-TDP43 Δ NLS axons.

Reviewer 3, Fig. 12 (Cropped from Fig.8 in the revised manuscript) – Quantification of the percent of contracting muscles in control and TDP43 Δ NLS co-culture with, or without the presence of G3BP1(190-208) peptides exclusively in the axonal/NMJ compartment.

6) Colocalization of transcripts with OPP puncta is not a reliable measure of its local translation, due to diffusion of labeled peptides (Enam et al., eLife 2020 “Puromycin reactivity does not accurately localize translation at the subcellular level”). A better approach would be proximity ligation assay (PLA) - puromylation to detect newly-synthesized molecules of a specific target protein by co-localization of a protein-specific antibody with an antibody recognizing puromycin.

Response – We agree with the reviewer regarding this issue. However, despite many attempts, we could not achieve a satisfactory puro-PLA staining of nuclear-encoded mitochondrial proteins. This is probably since Cox4i antibodies are mostly directed to the C-terminal region of the protein (as are multiple other antibodies targeting nuclear-encoded mitochondrial proteins), which can cause a problem when using this method (see Fig.2 in Dieck... Schuman, Nature Methods, 2015). Therefore, we chose an alternative approach to provide more satisfactory

evidence for the localized protein synthesis of Cox4i in axons. To this end, we used a recently published protocol for streptavidin pull-downs of biotin-OPP from sciatic nerve axoplasm (Terenzio et al, Science, 2018). Using this system, we were able to show that Cox4i is indeed locally synthesized within peripheral axons of sciatic nerves, and that its protein synthesis is attenuated by TDP-43 mislocalization (Fig.4p, Fig.S10b-c, Reviewer 3 Fig 6 above).

7) The co-localization of OPP puncta with mitochondria in Fig. 4H may be due to translation in mitochondria. OPP should also label mt-DNA encoded nascent peptides, but in a cycloheximide resistant manner.

Response – To examine if the OPP signal puncta within the mitochondria is achieved by labeling mt-DNA encoded nascent peptides, we conducted an additional experiment where we introduced cycloheximide to motor neuron axons prior and during OPP and mitoTracker staining. The colocalization analysis of OPP with mitochondria revealed that cycloheximide administration also decreased the degree of colocalization of OPP with mitochondria (Fig.S11a-b, attached below). This is in support of our original conclusion, that the reduction in OPP labeling colocalization with mitochondria in TDP43 Δ NLS axons is unrelated to aberrant labeling of OPP but to the reduction in mitochondria related protein synthesis.

Reviewer 3, Fig. 13 (cropped from fig. S11 in the revised manuscript) – Representative images and quantification of mitochondria colocalization with OPP in Cycloheximide (CHX)-treated versus untreated axons.

8) The manuscript Wang et al., J Neurochem. 2008 May;105(3):797-806. (“TDP-43, the signature protein of FTL-D, is a neuronal activity-responsive factor”) seems relevant and should be cited. This study found that TDP-43 is localized in the dendritic processing (P) body and it behaves as a translational repressor in an in vitro assay and repetitive stimuli by KCl enhanced the colocalization of TDP-43 granules with RNA-binding proteins known to regulate mRNA transport and local translation in neurons. While this study is not focused on axons, it first discusses a potential role for TDP-43 in repression of local translation in neurites.

Response – We thank the reviewer for bringing those interesting findings to our attention. The paper was added to our reference list.

9) RT-qPCR analysis of the TDP-43 bound mRNAs show a more than 6-fold increase in the binding of both Cox4i1 and ATP5A1 to TDP-43 (Fig.4f-g) in the diseased line.

What are the relative transcript levels of Cox4i1 and ATP5A1 transcript in ALS and control iPS-MNs?

Response – To validate the total transcript levels, we performed RT-qPCR for ATP5A1 and Cox4i1 in C9ORF72 and control iPS-motor neuron cell-bodies, which revealed similar transcript levels. Additionally, to further support the findings of the TDP-43-RIP from C9ORF72 axons, added to the revised version of the manuscript, and the findings from TDP Δ NLS axons, we quantified smFISH for Cox4i1. This revealed a dramatic increase in the axonal levels of Cox4i1 mRNA, in accordance with our TDP-43-RIP results (Reviewer 3 Fig. 5 in this letter, see above).

Reviewer 3, Fig. 14 (Prepared for this response and not included in the revised version of the manuscript) – Relative mRNA expression of Cox4i1 and ATP5A1 in C9ORF72 and control iPS-motor neurons.

10) Supplementary Figure 5 indicated HB9 staining in MN axons. This seems odd, considering that HB9 is a nuclear transcription factor used as a marker for MN nuclei, not axons. (This is different for HB9-driven GFP expression, of course.)

Response – The figure (now Fig.S9A) indicates HB9-GFP driven expression and not HB9 staining, and therefore it is visible also in axons. The error in the figure was corrected.

11) Does axonal TDP-43 lead to mitochondrial dysfunction in axons? The manuscript provides no evidence for “synaptic mitochondria toxicity” as shown in the model.

Response – We thank the reviewer for this important comment. To further study the possibility of synaptic mitochondria toxicity, we utilized TDP Δ NLS-Thy1-MitoDendra mice. Using this mouse model, we were able to demonstrate extensive reduction in the volume of NMJ mitochondria following TDP-43 mislocalization in vivo (Fig.5g-h in the revised manuscript, Reviewer 3 Fig 7 above). Furthermore, we also observed a reduction in pre-synaptic TMRE levels in vitro, marking a general alteration in NMJ pre-synaptic mitochondria function (Fig.5i-j in the revised manuscript). As mentioned earlier in the response to this reviewer (Reviewer 3 Fig. 11 above), application of G3BP1 peptides exclusively to the axonal/NMJ compartment in these co-cultures enabled recovery of synaptic mitochondria activity. Thus, we conclude that TDP-43 mislocalization and the subsequent events cause toxicity to NMJ mitochondria.

12) Suppl. Fig. 1B. The finding of TDP-43 loss from the nucleus and mislocalization in ALS patient-derived iPSCs into the cytoplasm is surprising, since a slight increase in soluble cytoplasmic TDP-43 is more typically observed in the field. Is this 1 shown cell representative? There is no quantification for this phenomenon shown. The high abundance of TDP-43 outside the nucleus even in control cells raises concerns about the specificity of the staining.

Response – We thank the reviewer for bringing this subject to our attention. To better understand the mislocalization of TDP-43 in our iPSC-derived motor neurons, we repeated the previous experiment to provide additional representative pictures of TDP-43. As our new images show, most of the TDP-43 in C9ORF72 iPSC-motor neurons is still contained within the nucleus. However, the levels of cytoplasmic TDP-43 are increased (Fig.S1b, attached below). Additionally, we performed staining of pTDP-43 together with G3BP1 in iPSC motor neurons cell bodies and proximal axons. This experiment revealed that the colocalization of pTDP-43 with G3BP1 is increased in mutant neurons (Fig.S3 and Fig.S4 in the revised manuscript). Our new results further strengthen our hypothesis that TDP-43 is mislocalized to the cytoplasm in human iPSC-motor neurons from ALS patients, and support the idea of TDP-43 gain of function, as TDP-43 still appears at the cell nucleus.

Reviewer 3, Fig. 15 (cropped from fig. S1 in the current manuscript) – Representative images demonstrating TDP-43 cytoplasmic mislocalization in C9ORF72 versus control iPSC-motor neurons. Dashed line indicates cell nucleus (DAPI).

Reviewer 3, Fig. 16 (Cropped from Fig. S3 in revised manuscript with relevant experiment) – a-d) Representative images (a) and quantitative analysis of pTDP-43-G3BP1 colocalization (b), G3BP1 intensity (c), and pTDP-43 intensity (d) in cell bodies of TDP43 Δ NLS motor neurons. e-f) Representative images and quantitative analysis of pTDP-43-G3BP1 colocalization in proximal axons of TDP43 Δ NLS motor neurons.

Reviewer 3, Fig. 17 (Cropped from Fig. S4 in revised manuscript with relevant experiment) – a-d) Representative images (a) and quantitative analysis of pTDP-43-G3BP1 colocalization (b), G3BP1 intensity (c), and pTDP-43 intensity (d) in cell bodies of C9ORF72 iPS-motor neurons. e-f) Representative images and quantitative analysis of pTDP-43-G3BP1 colocalization in proximal axons of C9ORF72 iPS-motor neurons.

Minor points:

Capitalized letters (A,B,C...) are used in figures but non-capitalized in figure legend (a,b,c...).

Response – This was corrected in the text and figures. They now show non-capitalized letters, with accordance to the journal guidelines.

Fig. 1N/O only show quantification of axonal TDP-43. How much are TDP-43 levels increased in the soma?

Response – We added a new graph quantifying the change in somatic levels of TDP 43, which are also significantly increased (Fig.S2e).

Reviewer 3, Fig. 18 (cropped from fig S2 in revised manuscript) – Western blot analysis of TDP-43 levels in cell-bodies of TDPΔNLS motor neurons.

Presumably tTDP43 and tERK in Fig. 1 stands for “total” proteins but is not explained in the legend or text.

Response – The error was corrected. Now total TDP-43 is marked as just TDP-43, and tERK meaning is explained in the legend as ERK1/2 (total ERK levels).

“b, d) Quantification of TDP-43(c) or pTDP-43(e) signal within” should be “TDP-43(b) or pTDP-43(d)”

Response – The error was corrected.

Size bars in H and J are indicated as 10um but seem at different magnifications.

Response - The error was corrected, the scale bar indicated in the updated Fig. 1h legend is 20µm.

Figure 7 is less than clear and even with its legend hard to understand.

Response – We changed the model to be simpler and more self-explanatory.

Reviewer 3, Fig. 19 (cropped from original fig. 8) - Axonal mislocalization of TDP-43 and its phosphorylated form create G3BP1-positive RNP condensates, which decrease local translation of nuclear-encoded mitochondrial proteins by sequestering their mRNA. This results in lack of nuclear-encoded mitochondrial proteins which leads to mitochondria toxicity and NMJ degeneration.

We thank the reviewers and hope they will agree that the revisions strengthen our paper, and the revised study is now suitable for publication in Nature Communications.

REVIEWER COMMENTS

Reviewer #1 (Remarks to the Author):

The authors have submitted a thoroughly revised manuscript with new data and clarifications that address previous comments from both reviewers and is now suitable for publication.

Reviewer #3 (Remarks to the Author):

This revised manuscript provides a wealth of important novel findings that link axonal mislocalization of TDP-43 to G3BP1-positive RNA-condensate assembly, thus inhibiting local mitochondrial protein synthesis in distal axons and NMJs. It thus provides very strong support to the idea of a toxic gain of function of cytoplasmic / axonal TDP-43 contributing to the disease process. It also provides exciting data on targeting G3BP1 granules to restored local translation and rescued TDP-43-derived toxicity in both axons and the NMJ.

I thank the authors for their extensive effort to respond to the reviewers' comments and they have addressed all my concerns adequately. I believe that it has become a much stronger manuscript for publication in this journal.

There are a few remaining issues that the authors should address before publication:

- 1) Most importantly, the authors have prepared multiple figures just for reviewers but not included in the manuscript or supplement, which seems an awfully wasted effort. These points were raised to strengthen the manuscript, not to make the reviewer happy.
E.g. figure "Reviewer 3, Fig. 1." addresses an important point but is neither mentioned nor included in the manuscript or supplement. The same is true for "Reviewer 3, Fig 5. (prepared for this response and not included in the current manuscript)" and "Reviewer 3, Fig 8" and "Reviewer 3, Fig. 14".
- 2) Abstract: "In axons, TDP-43 is hyper-phosphorylated and promotes G3BP1-positive RNA-condensate assembly, consequently inhibiting local protein synthesis in distal axons and NMJs." With the authors use "RNP-condensates" throughout the text, using "RNA-condensates" in the abstract may be confusing.
- 3) Another remaining issue I would like the authors to address is the unnecessary overstatement at the end of the discussion. There is no evidence that "regulating axonal levels of TDP-43" alone would "reverse the disease outcome". While "the TDP-43 cytoplasmic-related pathology could be reversed by ceasing TDP-43 Δ NLS expression", adding doxycycline to the mouse model reverses all kind of TDP-43 Δ eNLS pathologies, including loss of endogenous nuclear TDP-43 (Walker et al. 2015). It appears far more plausible that a more general strategy to reverse cytoplasmic TDP-43 mislocalization would be therapeutic, instead of targeting axons only.
- 4) Very minor random finding: author spelling in reference #19. "Holzbaurb, E. L. F."

Point by point answer to the reviewer comments

Response to Reviewer #1 & #2

The authors have submitted a thoroughly revised manuscript with new data and clarifications that address previous comments from both reviewers and is now suitable for publication

Response – We thank the reviewers for their kind words and advice that substantially improved the manuscript.

Response to Reviewer #3

This revised manuscript provides a wealth of important novel findings that link axonal mislocalization of TDP-43 to G3BP1-positive RNA-condensate assembly, thus inhibiting local mitochondrial protein synthesis in distal axons and NMJs. It thus provides very strong support to the idea of a toxic gain of function of cytoplasmic / axonal TDP-43 contributing to the disease process. It also provides exciting data on targeting G3BP1 granules to restored local translation and rescued TDP-43-derived toxicity in both axons and the NMJ.

I thank the authors for their extensive effort to respond to the reviewers' comments and they have addressed all my concerns adequately. I believe that it has become a much stronger manuscript for publication in this journal.

Response – We thank this reviewer for his helpful comments that aided our manuscript.

There are a few remaining issues that the authors should address before publication:

1) Most importantly, the authors have prepared multiple figures just for reviewers but not included in the manuscript or supplement, which seems an awfully wasted effort. These points were raised to strengthen the manuscript, not to make the reviewer happy.

E.g. figure "Reviewer 3, Fig. 1." addresses an important point but is neither mentioned nor included in the manuscript or supplement. The same is true for "Reviewer 3, Fig 5. (prepared for this response and not included in the current manuscript)" and "Reviewer 3, Fig 8" and "Reviewer 3, Fig. 14".

Response – We agree with the reviewer on this comment. We added the figures and updated our main text as well as figure legends accordingly.

"Reviewer 3, Fig. 1." is now Fig.S3m-p.

"Reviewer 3, Fig 5" is now Fig.S10d.

"Reviewer 3, Fig 8" is now Fig.S3q-r.

"Reviewer 3, Fig. 14" is now Fig.S10b-c.

2) Abstract: “In axons, TDP-43 is hyper-phosphorylated and promotes G3BP1-positive RNA-condensate assembly, consequently inhibiting local protein synthesis in distal axons and NMJs.” With the authors use “RNP-condensates” throughout the text, using “RNA-condensates” in the abstract may be confusing.

Response – We thank the reviewer for noticing this mistake. We corrected it.

3) Another remaining issue I would like the authors to address is the unnecessary overstatement at the end of the discussion. There is no evidence that “regulating axonal levels of TDP-43” alone would “reverse the disease outcome”. While “the TDP-43 cytoplasmic-related pathology could be reversed by ceasing TDP-43 Δ NLS expression”, adding doxycycline to the mouse model reverses all kind of TDP-43delNLS pathologies, including loss of endogenous nuclear TDP-43 (Walker et al. 2015). It appears far more plausible that a more general strategy to reverse cytoplasmic TDP-43 mislocalization would be therapeutic, instead of targeting axons only.

Response – The text was edited to prevent this overstatement.

4) Very minor random finding: author spelling in reference #19. “Holzbaurb, E. L. F.”

Response – The text was edited.